

# Constraining braneworlds with entanglement entropy

Hao Geng[1★], Andreas Karch[2†], Carlos Perez-Pardavila[2‡], Lisa Randall[1∘],
Marcos Riojas[2§], Sanjit Shashi[2¶] and Merna Youssef[2‖]

**1** Jefferson Laboratory, Department of Physics, Harvard University,
17 Oxford St., Cambridge, MA 02138, USA
**2** Theory Group, Weinberg Institute, Department of Physics,
University of Texas, 2515 Speedway, Austin, TX 78712, USA

★ haogeng@fas.harvard.edu , † karcha@utexas.edu , ‡ cjp3247@utexas.edu ,
∘ randall@g.harvard.edu , § marcos.riojas@utexas.edu ,
¶ sshashi@utexas.edu , ‖ myoussef@utexas.edu

## Abstract

We propose swampland criteria for braneworlds viewed as effective field theories of defects coupled to semiclassical gravity. We do this by exploiting their holographic interpretation. We focus on general features of entanglement entropies and their holographic calculations. Entropies have to be positive. Furthermore, causality imposes certain constraints on the surfaces that are used holographically to compute them, most notably a property known as causal wedge inclusion. As a test case, we explicitly constrain the Dvali–Gabadadze–Porrati term as a second-order-in-derivatives correction to the Randall–Sundrum action. We conclude by discussing the implications of these criteria for the question on whether entanglement islands in theories with massless gravitons are possible in Karch–Randall braneworlds.



# 1   Introduction

Braneworld models of the universe as being embedded in a higher-dimensional spacetime have led to a plethora of insights into particle physics and quantum gravity [1]. The original scenario of Randall and Sundrum [2] considers $(d+1)$-dimensional[1] "bulk" Einstein gravity with Newton constant $G_{d+1}$, cosmological constant $\Lambda = -\frac{d(d-1)}{2L^2}$ (with $L$ a length scale), and two $d$-dimensional branes, both with action proportional to their worldvolumes. The Randall–Sundrum (RS) brane actions take the schematic form

$$I_{\mathrm{RS}} = -T \int_{\mathrm{brane}} \sqrt{-\tilde{g}} \, . \tag{1}$$

$T$ is a *tension* parameter for the brane with induced metric $\tilde{g}$. Furthermore, one may instead consider only one brane [3], which results in an infinite anti-de Sitter (AdS) bulk. While the action (1) does not directly arise in string theory, it can be thought of as a low-energy effective action for any brane-like object.[2] For a brane whose only degree of freedom is its geometric embedding, the action (1) is the only zero-derivative term consistent with symmetries, in particular reparameterization invariance.

Among its many phenomenological aspects (see for example [4–8]), the braneworld program provides a framework for studying quantum features of gravity on the branes [9, 10]. Furthermore, if the branes themselves have AdS geometry—in which case we call them Karch–Randall (KR) branes [11, 12]—then we can apply the AdS/CFT correspondence [13] to establish a *"doubly holographic"* principle under which the braneworld has an additional third description as a non-gravitating, strongly-coupled conformal field theory. Thus, we can apply the usual tools of holography relating the classical bulk to the CFT in order to study semiclassical gravity on the branes indirectly while circumventing several, more complicated calculations. As discussed below, one recent application of double holography has been to tackle the black-hole information problem beyond 2d gravity (as in [14]).

We can go beyond the RS term. In principle, the branes are coarse descriptions of warped compactifications in string theory, since, as mentioned above, the RS Lagrangian (1) is only the leading-order possible term in a derivative expansion [15, 16].[3] The action could have higher order corrections that go beyond the RS term to include higher-derivative interactions.

---

[1]Technically, [2] takes $d = 4$ to reflect our universe, but we keep $d$ general for now, only requiring $d > 2$.

[2]By "brane-like object," we mean any $(d+1)$-dimensional source of energy density in the bulk whose size in one spatial direction is much smaller than the AdS scale $L$ and in other directions is of $O(L)$. For such an object, we can use the low-energy effective action in its derivative expansion to describe its dynamics.

[3]We should be clear that not all holographic boundary conditions are described by brane-like objects, as discussed by [17, 18]. For example, the Janus solutions to type IIB supergravity [19–23] describe a class of smooth geometries dual to conformal defect systems. These situations are simply described by what we referred to as a brane-like object above and so are not a part of our analysis.

One such example is the Dvali-Gabadadze-Porrati (DGP) [24, 25] term

$$I_{\mathrm{DGP}} = \frac{1}{16\pi G_{\mathrm{b}}} \int_{\mathrm{brane}} \sqrt{-\tilde{g}}\tilde{R}, \qquad (2)$$

where $\tilde{R}$ is the Ricci scalar computed from $\tilde{g}$ and $G_{\mathrm{b}}$ is a new coupling. This is one of several two-derivative terms one can include as a correction to the RS action, following the standard logic of effective field theory (EFT) [26]—we expect "effective terms" in the Lagrangian that are higher order in the derivatives and come from the (assumed) UV completion. Furthermore, features of the UV completion in turn put constraints on what the coupling coefficients of these terms may be. Any choice of couplings in the EFT inconsistent with the existence of some UV completion describes a theory in the *swampland* [27–29].

Holography is intrinsic to the underlying string (or UV-complete) theory. Nonetheless, it is often assumed to persist to semiclassical regimes of quantum gravity [30]. If this is true then we may use the tools of holography to put swampland constraints on EFTs of gravity (cf. [31–35]). This observation has been used to constrain two-dimensional branes equipped with RS and Jackiw–Teitelboim terms [35]. In the same spirit, we apply this idea to the analogous brane-localized couplings [(1) and (2)] in higher ($d > 2$) dimensions by specifically focusing on aspects of "entanglement structures" (particularly Ryu–Takayanagi surfaces [36, 37]) constructed in the bulk spacetime and from some subregion of the dual CFT. Our goal is to put physical limitations on doubly holographic braneworlds using two criteria [38] based on the following considerations.

Entanglement structures are built from minimal, extremal codimension-2 bulk surfaces whose areas—including possible codimension-3 boundary terms—compute von Neumann entropies of CFT subregions. On quantum-theoretic grounds, such entropies must be positive. However, certain ranges of brane-localized couplings induce negative entropies through the codimension-3 boundary terms. Such couplings are in the swampland.

Additionally, we can look to the relationship between entanglement structures and causal structures. Both entanglement and causality can be used to devise proposals for a bulk region associated with the degrees of freedom on a CFT subregion $\mathcal{R}$ [39–43]. Fundamental causality conditions in the field theory imply that the resulting bulk "causal wedge" associated with $\mathcal{R}$ must always be included in the analogous "entanglement wedge" of $\mathcal{R}$—a property called *causal wedge inclusion (CWI)*. And so, we employ CWI as a swampland criterion on brane-localized couplings in braneworld theories.

We should say that using CWI as a swampland condition (but without branes) is not new. It has also been used to constrain Gauss–Bonnet gravity [33] and explored in Einstein cubic gravity [44]. However, causality in these "higher-derivative" theories need not be the same as in Einstein gravity, in that the fastest causal modes may not be lightlike but rather superluminal [45, 46], so causal structures often need to be reformulated. Fortunately, this is not a problem if we only have higher-derivative terms localized to the brane [47, 48].

## 1.1 Relevance to black-hole information

In recent years, doubly holographic braneworld models like those studied in this paper have served as tools by which to probe the problem of black-hole information in $d > 2$ dimensions [14, 49]. Indeed, one of our motivations for constraining such models is to rein in their application, so that we are not led to artificial conclusions by unphysical setups.

Let us review the recent progress of black-hole information in AdS. The basic setup of [50, 51] is to couple a gravitating black hole to a non-gravitating thermal bath. We work in a semiclassical regime with quantum fields turned on but not backreacting onto the metric. Hawking radiation is collected in a subregion $\mathcal{R}$ on some Cauchy slice of the bath. As we want

to include quantum effects, the entropy of this Hawking radiation $S[\mathcal{R}]$ is computed by the *quantum extremal surface* prescription [52], which yields the "island rule" [53]:

$$S[\mathcal{R}] = \min_{\mathcal{I}} \text{ext} \left( S_{\text{scl}}[\mathcal{R} \cup \mathcal{I}] + \frac{A[\partial \mathcal{I}]}{4G_{\text{N}}} \right). \tag{3}$$

$\mathcal{I}$ is a bulk region that is disconnected from $\mathcal{R}$, and so we call it an "island." $S_{\text{scl}}[\mathcal{R} \cup \mathcal{I}]$ is the entropy of matter fields in $\mathcal{R} \cup \mathcal{I}$, and $A[\partial \mathcal{I}]$ is the area of the boundary of the island. We need to find $\mathcal{I}$ such that the expression in parentheses, which is a "generalized entropy," is minimized. For both evaporating [50, 51] and eternal [54] 2d black holes, $S[\mathcal{R}]$ was found to follow the usual time-dependent Page curve [55]—thereby preventing an information paradox—thanks to the emergence of an island (i.e. a transition from $\mathcal{I} = \varnothing$ to $\mathcal{I} \neq \varnothing$).

The higher-dimensional setup of [14] relies on treating the matter as holographic so that $S_{\text{scl}}$ can be computed using bulk geometry [56]. This is accomplished by embedding a single $d$-dimensional KR brane in a $(d + 1)$-dimensional AdS black geometry. However, [57] shortly after claimed that the existence of a time-dependent Page curve and an island may be due to the semiclassical theory on the brane being one of *massive* gravity, a feature long attributed to the presence of a non-gravitating bath [58, 59].

This aspect of the one-brane setup motivated the study of a two-brane setup embedded in bulk AdS spacetime [60]. Here, one of the branes acts as a gravitating bath, and so the semiclassical theory on the branes maintains a massless graviton in its spectrum [61]. To be concrete, we introduce the following terminology for the three different perspectives of the (two-KR-brane) braneworld afforded by double holography:

(I)  the bulk, which is classical $(d + 1)$-dimensional Einstein gravity on a "wedge" [62, 63] with $d$-dimensional KR branes (i.e. with AdS geometry);

(II) the intermediate picture, which is the effective field theory describing semiclassical $d$-dimensional (and notably, massless [61]) gravity on AdS; and

(III) the defect system, i.e. the $(d - 1)$-dimensional CFT on the tip of the bulk wedge.

Any version of the black hole information problem must be posed in the intermediate picture, in which we must explicitly account for quantum corrections. However, if there is a known bulk picture, then we can map the problem to an easier one of classical geometry in the bulk [56]. This is the power of double holography, as we will see in Section 2.

In [60], we studied two $d > 2$ dimensional, single-sided AdS-Schwarzschild black holes in the intermediate picture coupled to one another at infinity. However, unlike with a non-gravitating bath, the radiation region $\mathcal{R}$ (assumed to not be anchored to the defect) must be dynamically determined to protect diffeomorphism invariance, i.e. we minimize over both $\mathcal{I}$ and $\mathcal{R}$ in the island rule (3) to compute the entropy $S$.

With that in mind, one can consider several scenarios of how to divide the system in order to define an entanglement entropy. The most suitable for our purposes is to study in the fully quantum defect system (III) the von Neumann entropy of the defect in the thermofield double state (TFD). That is, we trace out the degrees of freedom of the double to obtain a density matrix for the CFT on the defect and calculate the associated entropy. In prescriptions (II) and (I) we are respectively looking either for a quantum extremal surface on the brane or, equivalently, a classical RT surface in the bulk that separates the two defects of the thermofield double state from each other.

In [60], we solved this particular problem using a bulk geometry with two $d$-dimensional KR branes [with the RS action (1)] embedded in a $(d + 1)$-dimensional AdS black string. We found that the entropy is computed by the bulk black string's horizon area, thus following a "trivial" (time-independent) Page curve rather than a "nontrivial" (time-dependent) one. The

Page curve's triviality in theories of massless gravity was earlier suggested by [57, 64] and asserted in [65, 66]. Furthermore unlike in the case with a non-gravitating bath, no entanglement island (in the sense of a disconnected bulk region) is formed.

Recent work [67–69] has claimed that one may get nontrivial Page curves and islands even with a gravitating bath (and thus, massless gravity) by turning on brane-localized couplings—particularly DGP terms (2)—treated as higher-derivative corrections to RS terms (1). Such a result requires that the coupling $G_b$ in (2) be negative on at least one of the branes, but this alone is not necessarily an unphysical assumption for AdS branes. However, in such theories, one may get entropies that are smaller than Bekenstein–Hawking or even time-dependent. Such answers violate *bulk* causal wedge inclusion, putting such scenarios in the swampland. In other words, the radiation entropy for UV-consistent couplings must be the bulk Bekenstein–Hawking value at all times.

## 1.2 Outline

In Section 2, we will set the stage by reviewing the holography of the two-brane setup, in particular discussing how the branes modify the usual entanglement entropy prescriptions [36, 37]. We will also justify both the positivity of entropy computed by the codimension-2 entanglement surfaces and the rationale for CWI.

In Section 3, we will scrutinize RS + DGP gravity [(1) + (2)] through both analytic and numerical methods. Instead of using $T$ and $G_b$ as our brane parameters, we find it more convenient to use the Euclidean angle of the AdS brane with the conformal boundary (the "brane angle" $\theta$) and a particular dimensionless combination of $G_b$ with the bulk Newton constant and curvature radius (the "DGP coupling" $\lambda$). The formulas that switch between these pairs of parameters are written in Section 3.1. The parameter space for two-brane systems is four-dimensional, so we explore how our swampland criteria exclude ranges of the DGP couplings for particular brane angles. See Figure 5 for the analytic constraints from entropy positivity and Figure 6 for the numerical points excluded by CWI.

In Section 4, we discuss the consequences of our exclusion criteria on the physics of two-brane braneworld models in the intermediate picture. In particular, we describe the implications for black-hole information in higher-dimensional ($d > 2$) KR branes with massless gravity. Essentially, our criteria—particularly CWI—mandate that UV-consistent two-brane setups furnish a trivial semiclassical Page curve for the entropy of Hawking radiation on the brane, in accordance with the ideas of [60, 64, 66], with the entropy always given by the horizon area. Thus, braneworld theories with massless gravity on the brane and whose entanglement structures yield nontrivial Page curves [67–69] contradict CWI and should be considered in the swampland.

## 2 The holography of two-brane models

We first review the holographic interpretations of the two-KR-brane setups with just RS terms so as to lay the groundwork for applying the tools of holography to more exotic braneworld constructions.

We start by recalling the setup in question. It consists a bulk spacetime $\mathcal{M}$ with two codimension-1 boundaries $\mathcal{Q}_1$ and $\mathcal{Q}_2$. The action (including Gibbons–Hawking terms) is

$$I_{\text{RS1}} = \frac{1}{16\pi G_{d+1}} \int_{\mathcal{M}} \sqrt{-g} \left[ R + \frac{d(d-1)}{L^2} \right] + \sum_{i=1}^{2} \frac{1}{8\pi G_{d+1}} \int_{\mathcal{Q}_i} \sqrt{-\tilde{g}_i}(K_i - 8\pi G_{d+1} T_i), \quad (4)$$

where $\tilde{g}_i$ is the metric of $\mathcal{Q}_i$ and $K_i$ is the trace of the extrinsic curvature $K_{i,\mu\nu} = \nabla_\mu n_{i,\nu}$ of

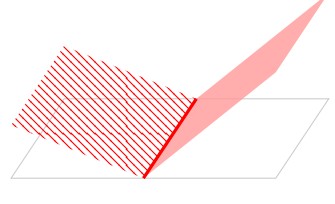

(a) $(d+1)$-dimensional bulk

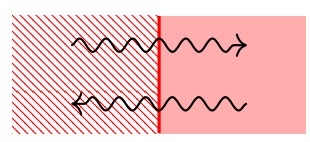

(b) $d$-dimensional branes

Figure 1: A schematic representation of the two-brane setup of interest in this paper. (a) is the classical bulk configuration consisting of two KR branes (in red) embedded in an ambient $\mathrm{AdS}_{d+1}$ spacetime. (b) is the brane-localized "intermediate picture" of two interacting $\mathrm{AdS}_d$ universes coupled together along a $(d-1)$-dimensional interface (the "defect"). Both pictures can be seen as dual to a $(d-1)$-dimensional CFT on this defect.

$\mathcal{Q}_i$ (with $n_{i,\nu}$ being the unit normal and the $\mu, \nu$ indices projected onto $\tilde{g}_i$). For now, we have suppressed the coordinate dependence to simplify the notation. Additionally, $i = 1, 2$ labels the two branes while Greek letters are spacetime indices. The equations of motion of this action are

$$G_{\mu\nu} = \frac{d(d-1)}{2L^2} g_{\mu\nu}, \tag{5}$$

$$K_{i,\mu\nu} = (K_i - 8\pi G_{d+1} T_i)\tilde{g}_{i,\mu\nu}. \tag{6}$$

(5) implies that the bulk geometry is locally $\mathrm{AdS}_{d+1}$. (6) contracted with $\tilde{g}_i$ yields

$$K_i = \frac{8\pi G_{d+1} d}{d-1} T_i. \tag{7}$$

In other words, solutions to (6) are constant-curvature boundaries in $\mathrm{AdS}_{d+1}$.

We are interested in branes that have timelike boundary and thus extend eternally in the bulk. This is only the case when the tensions are "subcritical" [11, 70] or

$$|T_i| < \frac{d-1}{8\pi G_{d+1} L}. \tag{8}$$

For such tensions, the resulting brane has $\mathrm{AdS}_d$ geometry [11] and is thus deemed a Karch–Randall (KR) brane. Critical tensions saturating this bound yield Minkowski spacetime instead, and so the near-critical limit is the regime in which the brane's cosmological constant is small. This is often the limit of interest because it is the regime where the lowest-mass Kaluza–Klein (KK) mode of linearized fluctuations of the brane "localizes," yielding a bound graviton on the brane coupled to parametrically heavier CFT modes [11]. However, we will generally allow any subcritical tension and assume that there is some holographic description of the $(d+1)$-dimensional bulk theory as $d$-dimensional semiclassical gravity on the brane[4] and coupled to CFT fields [71]. We call this lower-dimensional theory the "intermediate picture" due to its role in double holography [56].

KR branes can be engineered by foliating $\mathrm{AdS}_{d+1}$ into $\mathrm{AdS}_d$ slices (cf. [72]). The individual foliates in such a slicing reach the conformal boundary. Taking just one of these slices as a KR brane yields a setup in which the intermediate picture describes a gravitating universe on $\mathrm{AdS}_d$ coupled at infinity to a non-gravitating CFT. When we instead construct a setup using two KR branes, they intersect along a $(d-1)$-dimensional surface (called the *defect*) at the conformal

---

[4]In this interpretation of the universe on the brane, the lowest-mass KK mode is still the graviton, but it has finite mass and is no longer parametrically lighter than the higher KK modes constituting the CFT.

boundary. In the intermediate picture, this defect is viewed as an interface between the two gravitating AdS$_d$ universes, and the full system has a massless graviton.[5] See Figure 1 for schematic representations of these dual setups.

So far, we have discussed the bulk perspective and the intermediate picture. However, the AdS/CFT correspondence [13] suggests that there is a third description—a conformal field theory on the defect (since it is the boundary of the bulk spacetime) [12]. The correspondence between the bulk and defect systems has more recently been dubbed "wedge holography" [62, 63][6] and falls under the broader umbrella of "double holography" equating the $(d + 1)$-dimensional bulk, $d$-dimensional intermediate, and $(d - 1)$-dimensional defect systems. The upshot is that physical quantities of the $(d-1)$-dimensional CFT are dual to various geometric data in the $(d + 1)$-dimensional bulk theory, and the dictionary translating between these two pictures can be used to further understand semiclassical physics in the intermediate picture without *explicitly* accounting for its quantum effects.

## 2.1 Entanglement entropy

The main entry of the holographic dictionary of relevance to our paper is the classical prescription for computing entanglement entropies of CFT subsystems. In AdS/CFT with no branes, the formula is given by the Ryu–Takayanagi (RT) prescription [36] (or its covariant extension of Hubeny–Rangamani–Takayanagi [37]); for a subregion $\mathcal{R}$ of the boundary CFT, the entanglement entropy to leading order in the limit $G_{d+1} \ll L^{d-1}$ is

$$S[\mathcal{R}] = \frac{1}{4G_{d+1}} \min_{\gamma_\mathcal{R} \sim \mathcal{R}} \text{ext}\, A[\gamma_\mathcal{R}]. \tag{9}$$

Here, $\gamma_\mathcal{R}$ is a $(d - 1)$-dimensional surface that is homologous to the CFT subregion $\mathcal{R}$, where "homology" means that there exists some codimension-1 bulk region $\Sigma$ such that

$$\partial\Sigma = \gamma_\mathcal{R} \cup \mathcal{R}, \qquad \partial\gamma_\mathcal{R} = \partial\mathcal{R}, \tag{10}$$

and $A[\gamma_\mathcal{R}]$ is the area functional for such surfaces. Essentially, this formula equates entanglement entropy with the area of the smallest extremal surface "anchored" to the CFT subregion $\mathcal{R}$ (Figure 2a). However, note that this does not include contributions from quantum fields in the bulk.

We can ask how this prescription is modified with two KR branes. The proposal [14, 60, 62, 63] is to equate the von Neumann entropy of the defect at some fixed time with a minimal extremal surface in the bulk via a modification of (9). Essentially, the homology condition (10) is modified such that the surface is anchored to the branes, rather than to the conformal boundary (Figure 2b). This generically introduces boundary terms in the entropy functional, and the extremization procedure also requires us to choose boundary conditions (dynamical or fixed) for the intersection of the surface with the branes. While a fixed boundary condition like Dirichlet is technically a mathematically consistent choice [77], imposing it amounts to partitioning degrees of freedom on the brane by hand, a procedure that we assert is physically incompatible with diffeomorphism invariance [60]. And so, we impose dynamical (e.g. Neumann) conditions on the RT surface as in [75].[7]

In the intermediate picture, this same entropy is interpreted as a generalized entropy that accounts not just for a leading-order semiclassical entropy, but also for quantum fields on

---

[5]Technically, the massless graviton exists in tandem with massive modes. This is called "bigravity" [61].

[6]It behooves us to mention that this logic also goes through for the configuration consisting of just one brane. There, the dual field theory is a BCFT [73, 74]. The relationship between this BCFT and the bulk theory has been dubbed the "AdS/BCFT correspondence" [75, 76].

[7]We should be clear that there is no proof in the sense of Lewkowycz–Maldacena [78, 79] for this prescription. However, there is evidence in the literature for this approach, e.g. in [25, 80].

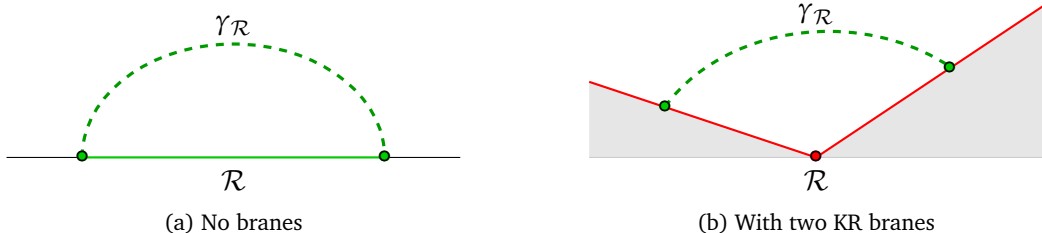

Figure 2: Schematic representations of the Ryu–Takayanagi surfaces for a fixed-time CFT subregion $\mathcal{R}$ in (a) AdS with no branes and (b) AdS with two KR branes. In (a), the surface is anchored to the boundary of $\mathcal{R}$ (i.e. $\partial \gamma_{\mathcal{R}} = \partial \mathcal{R}$); this is a Dirichlet condition on $\gamma_{\mathcal{R}}$. In (b), the CFT subregion is the $(d-1)$-dimensional defect at fixed time, and $\gamma_{\mathcal{R}}$ is anchored to the branes via a Neumann condition. We use Neumann for physical reasons; a Dirichlet condition would break diffeomorphism invariance on the brane.

the brane. Typically, the generalized entropy is calculated by a quantum extremal surface (QES) [52] rather than an RT surface, and the QES may even be the boundary of disconnected pieces of the entanglement wedge called "entanglement islands" [50, 51]. By the power of double holography, the generalized entropy in the intermediate picture can be computed by the area of a classical RT surface in the $(d+1)$-dimensional bulk [56]. This is the key feature of double holography that we will exploit in this paper.

In other words, the semiclassical entropy on the brane, which might normally be computed by employing the quantum extremal surface prescription, is encoded by a classical surface in the bulk. As classical surfaces are rather tractable to compute, double holography has allowed the use of one-brane setups to study the entropy of quantum fields in the presence of a gravitating black hole coupled to a non-gravitating thermal bath. Indeed, this has been the main approach to study higher-dimensional ($d > 2$) semiclassical gravity, starting with [14] and extended by [25, 57, 81] and many others as part of the entanglement island program.

While we will briefly discuss black-hole information and islands, our primary focus is on how the bulk entropy prescription can be used to constrain brane-localized couplings added by hand to the theory. In other words, we are assuming the holographic dictionary to be a fundamental aspect of UV physics that is capable of providing swampland criteria on effective field theories of gravity (cf. [31–34]).

We now briefly justify the two main swampland criteria rooted in entanglement. We will later explain how we concretely use them in Section 2.2.

**Positivity of entropy**   One immediate swampland criterion is the positivity of entropy. Generally, microcanonical entropy is a logarithmic count of the microstates $\Omega$ in a system,

$$S \sim \log \Omega, \quad \Omega \geq 1. \tag{11}$$

Entanglement entropy (i.e. the von Neumann entropy of a reduced density matrix) can also be written as (11), albeit with $\Omega$ being real rather than an integer. By employing a Schmidt decomposition of the initial state, we can identify $\Omega$ with a function of Schmidt coefficients that is necessarily greater than 1,[8] and this crucially lets us insist that entanglement entropy be positive.

---

[8]To be concrete, suppose that we have a normalized state $|\Psi\rangle \in \mathcal{H}_A \otimes \mathcal{H}_B$. For simplicity, if we take these Hilbert spaces to be finite-dimensional, we can define a finite $n \equiv \min(\dim \mathcal{H}_A, \dim \mathcal{H}_B)$. We then Schmidt-decompose $|\Psi\rangle = \sum_{i=1}^{n} \alpha_i |a_i\rangle \otimes |b_i\rangle$, where $\{|a_i\rangle\} \subset \mathcal{H}_A$ and $\{|b_i\rangle\} \subset \mathcal{H}_B$ are orthonormal sets and each $\alpha_i$ lies between 0 and 1 (and $\sum_{i=1}^{n} \alpha_i^2 = 1$). Pushing through the von Neumann entropy formula yields $\Omega \equiv \prod_{i=1}^{n} \alpha_i^{-2\alpha_i^2}$, which is easily seen to be $\geq 1$.

However, the entanglement entropy of a subregion formed by splitting a connected interval is typically UV-divergent in QFT [82]. This is reflected in the RT prescription with no branes present through the fact that RT surfaces reach the conformal boundary, thus formally being infinite in area. Nonetheless, these "absolute" entropies are still positive. While we can subtract away the infinity through some renormalization scheme and get a negative answer, such a result does not contradict the positivity of the "bare" entropy.

But in two-brane configurations, we no longer have this positive divergence in the area. This is because we are not splitting any CFT regions. Furthermore, the full entropy functional picks up "boundary-term" contributions from the couplings on the branes, and so it is *a priori* possible to have a functional with negative values. However, a negative-entropy UV-finite RT surface contradicts (11), and so couplings giving such a result would be in the swampland.

**The basis of CWI** Another swampland condition coming from AdS/CFT is causal wedge inclusion (CWI)—the property that the entanglement wedge always contains the causal wedge. Let us discuss why such a criterion is fundamental.

First, we recall the motivations and definitions of the causal and entanglement wedges. Both ideas came out of attempts [39–43] to answer to the question, "What is the holographic dual of the reduced density matrix $\rho_{\mathcal{R}}$ on the CFT subregion $\mathcal{R}$?" Recall that this reduced density matrix is defined by tracing out complementary degrees of freedom $\mathcal{R}^c$ and encodes the degrees of freedom of $\mathcal{R}$:

$$\rho_{\mathcal{R}} \equiv \text{Tr}_{\mathcal{R}^c}(\rho)\,, \quad S[\mathcal{R}] \equiv -\text{Tr}_{\mathcal{R}}(\rho_{\mathcal{R}} \log \rho_{\mathcal{R}})\,. \tag{12}$$

Starting with $\mathcal{R}$, we define its future domain of dependence $D_+[\mathcal{R}]$ as the points $p$ for which all past-directed causal curves starting at $p$ intersect $\mathcal{R}$. We analogously define the past domain of dependence $D_-[\mathcal{R}]$ as the points from which all future-directed causal curves intersect $\mathcal{R}$. The union $D[\mathcal{R}] = D_+[\mathcal{R}] \cup D_-[\mathcal{R}]$ is the full domain of dependence and is shaped like a diamond.

The causal wedge $\text{CW}[\mathcal{R}]$ is defined in terms of the domain of dependence in the boundary system $D[\mathcal{R}]$. Basically, we take all bulk causal curves which start in $D_-[\mathcal{R}]$ and end in $D_+[\mathcal{R}]$ [40]. If $\mathcal{R}$ is a fixed-time subregion (say at $t = 0$), this comes with a codimension-2 surface $\xi_{\mathcal{R}}$ that is anchored to $\mathcal{R}$ and bounds the $t = 0$ cross-section of the causal wedge.

Meanwhile, the entanglement wedge $\text{EW}[\mathcal{R}]$ is defined in terms of the RT surface $\gamma_{\mathcal{R}}$ of $\mathcal{R}$. Suppose that $\mathcal{R}$ is spacelike. We then take the bulk codimension-1 "homology surface" bounded by $\mathcal{R}$ and $\gamma_{\mathcal{R}}$ within the Cauchy slice containing $\mathcal{R}$ [i.e. $\Sigma$ in (10)] and define $\text{EW}[\mathcal{R}]$ as the domain of dependence of this homology surface [43]. Just like $\text{CW}[\mathcal{R}]$, the entanglement wedge asymptotes to $D[\mathcal{R}]$.[9] However, $\text{EW}[\mathcal{R}]$ is constructed from entanglement structure and is generally distinct from $\text{CW}[\mathcal{R}]$. In particular, for $\mathcal{R}$ at $t = 0$, the corresponding RT surface $\gamma_{\mathcal{R}}$ need not be the same as $\xi_{\mathcal{R}}$.

In fact, one can argue that the entanglement wedge is generally supposed to reach further into the bulk than the causal wedge [43]. This is precisely the statement of CWI and is depicted in Figure 3. Furthermore, for a boundary subregion $\mathcal{R}$ at $t = 0$, CWI implies a condition on the codimension-2 surfaces $\xi_{\mathcal{R}}$ and $\gamma_{\mathcal{R}}$ residing on the bulk $t = 0$ Cauchy slice—the region bounded by $\gamma_{\mathcal{R}}$ must contain the region bounded by $\xi_{\mathcal{R}}$. Physically, CWI states that the RT surface of $\mathcal{R}$ is generally causally disconnected from the domain of dependence $D[\mathcal{R}]$, at best only being accessible via null signals, and the argument relies on causality in the boundary theory.

Let us give the intuition behind this argument; for a more thorough proof, see [43]. Again, take $\mathcal{R}$ to be a CFT subregion on the $t = 0$ Cauchy slice. Suppose that the RT surface $\gamma_{\mathcal{R}}$

---

[9]This is because the bulk and boundary causal structures are compatible with one another (cf. [47,48]), and so null rays confined to the boundary can also be seen as bulk null rays "at infinity".

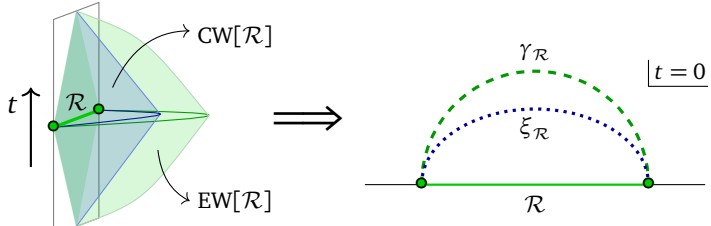

Figure 3: On the left, a cartoon of causal wedge inclusion (CWI) for a CFT subregion $\mathcal{R}$ at fixed boundary time $t = 0$. The inner (blue) wedge is the causal wedge $CW[\mathcal{R}]$, while the outer (green) wedge is the entanglement wedge $EW[\mathcal{R}]$. We can further consider the cross-section of this picture with the initial bulk Cauchy slice (right), in which $\gamma_{\mathcal{R}}$ (green, dashed) is the RT surface wrapping around $EW[\mathcal{R}]$ while $\xi_{\mathcal{R}}$ (blue, dotted) is the codimension-2 surface wrapping around $CW[\mathcal{R}]$. CWI implies that the region on this Cauchy slice bounded by $\xi_{\mathcal{R}}$ must be within the analogous region bounded by $\gamma_{\mathcal{R}}$.

is located between $\xi_{\mathcal{R}}$ and $\mathcal{R}$ on this slice. This implies that the RT surface can be reached causally by timelike signals sent from the domain of dependence $D[\mathcal{R}]$.

Such timelike signals can be realized in the field theory as small perturbations of the Hamiltonian that evolve the reduced state on $\mathcal{R}$. The perturbed state is

$$\rho_{\mathcal{R}'} = U\rho_{\mathcal{R}}U^{\dagger}, \tag{13}$$

where $U$ is some unitary with support on just $D[\mathcal{R}]$.

If $\gamma_{\mathcal{R}}$ is in the causal wedge, there exists a perturbation which corresponds to a deformation of the bulk metric in the vicinity of $\gamma_{\mathcal{R}}$. Thus, applying this perturbation would alter the area of $\gamma_{\mathcal{R}}$ and change the entropy. However, on the CFT side, we can use the invariance of the partial trace under change of basis (since the Hilbert spaces on $\mathcal{R}$ and $\mathcal{R}'$ are isomorphic) to equate the $n$th Rényi entropy of the perturbed state to that of the original state:

$$S_n[\rho_{\mathcal{R}'}] = -\frac{1}{n-1}\log\text{Tr}_{\mathcal{R}'}[(\rho_{\mathcal{R}'})^n] = -\frac{1}{n-1}\log\text{Tr}_{\mathcal{R}}[(\rho_{\mathcal{R}})^n] = S_n[\rho_{\mathcal{R}}]. \tag{14}$$

Recall that the entanglement entropy is the $n \to 1$ limit of the Rényi entropy. So, the entanglement entropy should *not* change under the perturbation. This is a contradiction and we thus conclude that $\gamma_{\mathcal{R}}$ cannot be strictly within the causal wedge of $\mathcal{R}$, after all.

Note that the arguments of [43] and the discussion above are set in the realm of AdS/CFT without KR branes. However, we expect CWI to also hold with branes. This is because all that is necessary for CWI is the notion of a causal wedge and an entanglement wedge. Just having a working proposal for both is enough. More specifically, the causal wedge in a two-brane system can again be constructed from causal curves. Meanwhile, we consider the entanglement wedge to be constructed from the surface found via the modified entanglement prescription outlined in the previous exposition of Ryu–Takayanagi.

## 2.2 Black spacetimes as a testing ground

As one application of entropy positivity and CWI as swampland constraints on higher-derivative corrections to the RS action, we examine two-KR-brane systems embedded in eternal two-sided "black" spacetimes in the bulk. All of our analysis will take place on one side of the initial-time slice. In principle, we may consider other two-brane setups in other geometries, and doing so may yield additional nontrivial constraints on brane couplings. We focus on black spacetimes because they have particularly useful entanglement and causal features.

**Positivity of horizon entropy**  The original motivation of the RT prescription was the Bekenstein–Hawking identification of a black-hole horizon's area with the black hole's entropy [49, 83], which was also found in string theory [84]. Indeed, RT is meant to be a generalization of the Bekenstein–Hawking formula. We recover the latter when we consider a CFT in a thermal state, which is realized holographically by a one-sided black hole. When $\mathcal{R}$ is taken to be the full boundary, then the associated RT surface is the horizon.

In two-brane setups embedded in black holes, the entropy functional is generically modified by boundary terms coming from brane-localized couplings. Nonetheless, the horizon is still expected to be an extremal surface. Given this, the quantity $A_h$ (defined as the sum of the "bulk" horizon area with its boundary terms at the branes) must then at least upper-bound the minimum of the entropy functional $S_{min}$, which represents entropy in the semiclassical limit and is thus positive. Thus,

$$S_h \equiv \frac{A_h}{4G_N} > S_{min} > 0 \,. \tag{15}$$

A negative value of $S_h$ (induced by sufficiently negative boundary terms in the functional) contradicts the positivity of entropy. However, we might mathematically get such a result from the brane-localized couplings, but such theories live in the swampland.

**CWI in black spacetimes**  Consider one side of a two-sided eternal black geometry at some initial Cauchy slice $t = 0$. We take $\mathcal{R}$ to be the boundary of the chosen side at this slice. In this case, $\mathcal{D}[\mathcal{R}]$ is the full boundary, with the earliest and latest points respectively being past and future timelike infinity. The bulk causal wedge $CW[\mathcal{R}]$ can then be constructed by shooting null rays from these points. Upon doing so, we see that the $t = 0$ slice of $CW[\mathcal{R}]$ is precisely the exterior region outside of the horizon,[10] and so the $t = 0$ bounding surface $\xi_{\mathcal{R}}$ of the causal wedge is itself the horizon.

The point is that applying the CWI criterion in black geometries simply becomes a matter of explicitly computing the RT surface $\gamma_{\mathcal{R}}$ with $\mathcal{R}$ being the boundary system at $t = 0$. For a general braneworld with Einstein gravity in the bulk, $\gamma_{\mathcal{R}}$ will be either the horizon (in accordance with the Bekenstein–Hawking formula [49, 83]) or some other surface. The former is consistent with CWI, so the only point of concern is the latter case.

Fortunately, non-horizon surfaces can easily be seen to contradict CWI. In the maximally extended spacetime at $t = 0$, there is no interior geometry. Thus, the only conceivable non-horizon candidates for $\gamma_{\mathcal{R}}$ are those that intersect the horizon or those that reside completely in the exterior. In either case, there would be a part of the $t = 0$ slice of the exterior that is not a part of the entanglement wedge, which then implies $CW[\mathcal{R}] \not\subset EW[\mathcal{R}]$.

So, to summarize how to use CWI to potentially rule out a theory, we start by taking a two-KR-brane setup embedded in a black spacetime. We then search for $t = 0$ non-horizon extremal surfaces anchored to the branes. If we find one, we can compute its entropy $S_{ext}$ and compare it against that of the horizon $S_h$. *A priori* we have the following possibilities:

$$S_{ext} > S_h \implies \text{no violation of CWI}, \tag{16}$$

$$S_h > S_{ext} \implies \text{theory violates CWI}. \tag{17}$$

A theory with a non-horizon extremal surface for which the latter holds is in the swampland.

**Black string with RS terms**  A simple test is to check that both criteria are satisfied in two-KR-brane configurations solving the equations (5)–(6). We do not expect braneworld theories with only RS terms to reside in the swampland, since the bulk action is just Einstein gravity

---

[10]In fact, the causal wedge is the full exterior on the same side of the horizon as $\mathcal{R}$. One can see this by drawing the Penrose diagram and shooting null rays from past and future timelike infinity.

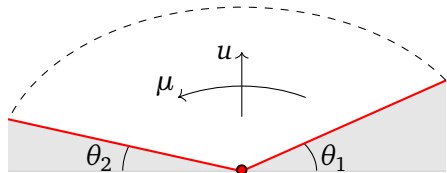

Figure 4: A fixed-$t$, fixed-$\vec{x}$ slice of a planar black string in AdS consisting of two planar KR branes (in red), each at a fixed angle $\theta_1, \theta_2 \in \left(0, \frac{\pi}{2}\right)$ with respect to the conformal boundary of the underlying $\text{AdS}_{d+1}$ spacetime. The defect (the red point) is at $u = 0$ and the horizon (the dashed line) is at $u = u_{\text{h}}$. The branes have constant extrinsic curvature, and so this geometry solves the equations of motion with the RS action (5)–(6).

while the brane action has only the terms corresponding to zeroth and first derivatives of the metric.[11] One black solution in which we can test this claim is the following $(d+1)$-dimensional planar AdS black-string geometry:

$$ds^2 = \frac{L^2}{u^2 \sin^2 \mu}\left[-h(u)dt^2 + \frac{du^2}{h(u)} + u^2 d\mu^2 + d\vec{x}^2\right], \quad h(u) = 1 - \frac{u^{d-1}}{u_{\text{h}}^{d-1}}. \tag{18}$$

Here, $t \in \mathbb{R}$, $u > 0$, and $\mu \in (0, \pi)$ are respectively the time, radial, and angular coordinates while $\vec{x} \in \mathbb{R}^{d-2}$ represents the $d-2$ remaining transverse, planar directions. The defect is located at $u = 0$, and $u = u_{\text{h}}$ is the horizon. The equation of motion (6) is solved by "planar KR branes" parameterized as

$$\mu = \theta_1, \quad \mu = \pi - \theta_2, \tag{19}$$

for constants $\theta_1, \theta_2 \in (0, \pi)$ with $\theta_1 + \theta_2 < \pi$, and taking $\mu \in (\theta_1, \pi - \theta_2)$. See Figure 4 for a visual depiction. Not only do these hypersurfaces each have constant extrinsic curvature, but the induced geometry of each is a planar AdS-Schwarzschild black hole. Both the tensions $T_1, T_2$ and the induced curvature radii $\ell_1, \ell_2$ of these branes are well-defined functions of the angles:

$$T_i = \frac{(d-1)}{8\pi G_{d+1} L} \cos \theta_i, \quad \ell_i = L \csc \theta_i. \tag{20}$$

In [60], we had explored the bulk entanglement surfaces in this two-KR-brane configuration. With just RS terms, the entropy functional has no boundary terms (see [25]), so we had found that the RT surface computing the entropy of the defect at $t = 0$ is always the horizon. Then, the entropy is computed by just the bulk area of the horizon:

$$S_{\text{h}}|_{\text{RS}} = \frac{L^{d-1}}{4 G_{d+1} u_{\text{h}}^{d-2}} \int_{\mathbb{R}^{d-2}} d\vec{x} \int_{\theta_1}^{\pi-\theta_2} \frac{d\mu}{\sin^{d-1}\mu}. \tag{21}$$

This is always positive. Furthermore, because the horizon is the RT surface, the entanglement wedge is precisely the causal wedge. Thus, there is no violation of CWI.

## 3 Constraining the DGP term

We can apply the criteria for entanglement surfaces in Section 2 to KR braneworlds beyond those with just RS terms. We do so in an explicit simple model, finding that violations are in-

---

[11]The zeroth-derivative term is the RS term and has one free parameter—the tension. The first-derivative term is the Gibbons–Hawking term proportional to extrinsic curvature, and its coefficient is fixed (relative to the bulk Einstein action's coupling) by requiring that the variational principle is well-defined.

deed mathematically possible and can be used to rule out particular brane-localized couplings on physical grounds.

The simplest higher-derivative term we can add to the RS action is one that is linear in the brane's intrinsic Ricci curvature. Following the conventions of [25, 85], we write the full action as

$$ I = \frac{1}{16\pi G_{d+1}} \int_{\mathcal{M}} \sqrt{-g} \left[ R + \frac{d(d-1)}{L^2} \right] + \sum_{i=1}^{2} \frac{1}{8\pi G_{d+1}} \int_{\mathcal{Q}_i} \sqrt{-\tilde{g}_i} \left[ K_i - 8\pi G_{d+1} T_i + \frac{G_{d+1}}{2G_{\mathrm{b},i}} \tilde{R}_i \right], \quad (22) $$

which has an additional term—the DGP term [24]—for each brane. The DGP Lagrangian is proportional to the Ricci scalar $\tilde{R}_i$ computed from the metric $\tilde{g}_i$. The $G_{\mathrm{b},i}$ couplings are new parameters each controlling the strength of this term on the associated brane.

We should mention that the original DGP model describes flat branes living in a flat bulk [24]. In the original model, there is no RS term and no bulk cosmological constant, and the gravitational coupling flows from $G_{\mathrm{b}}$ at high energies to $G_{d+1}$ at low energies. However, we are mainly interested with using the machinery of AdS/CFT to put theories in the swampland. To learn anything about flat branes in flat space, we may take flat-space limits of our AdS constraints. The punchline is that the the constraints from entanglement lose all power under flat-space limits, and that the only constraint we are left with is that $G_{\mathrm{b}} > 0$ in order to not have a wrong-sign Euclidean action with maxima instead of minima.[12] We will discuss these points in Appendix A (specifically, A.2).

In the current section, we first discuss the parameter space of two-KR-brane setups with RS + DGP terms, as well as how the entropy functional gets modified by DGP terms. We then compute the equations for extremal surfaces in the presence of these branes. What follows are our main results, i.e. the constraints put on this class of braneworld models, which we obtain via both an analytic approach and a numerical approach. Analytically, we will find that violations of CWI require at least one DGP coupling to be negative. We find further analytic lower-bounds on DGP couplings by requiring positivity of the horizon entropy. Last but not least we numerically find theories violating CWI leading to even more refined bounds.

## 3.1 Parametrizing KR branes with RS + DGP terms

Here we consider the DGP term as a higher-derivative correction to the RS term on a KR brane as in [25], so we are studying AdS branes in an AdS bulk. The brane-embedding equations are modified relative to (6) by adding terms that depend on Ricci curvature:

$$ K_{i,\mu\nu} = (K_i - 8\pi G_{d+1} T_i)\tilde{g}_{i,\mu\nu} - \frac{G_{d+1}}{G_{\mathrm{b},i}} \left[ \tilde{R}_{i,\mu\nu} - \frac{1}{2}\tilde{R}_i \tilde{g}_{i,\mu\nu} \right]. \quad (23) $$

However, the solutions turn out to be geometrically the same as without the DGP term, but with the tension parameter being redefined.[13] In terms of the angular parameterization (19), the brane tension of a KR brane with RS + DGP terms (derived in Appendix A) is

$$ T_i = \frac{d-1}{8\pi G_{d+1} L} \cos\theta_i - \frac{(d-1)(d-2)}{16\pi G_{\mathrm{b},i} L^2} \sin^2\theta_i. \quad (24) $$

We mention that the DGP term is one of three second-derivative terms we can add to the brane action. We can also add an action $\int \sqrt{-\tilde{g}_i}(a_i K_i^2 + b_i K_i^{\mu\nu} K_{i,\mu\nu})$ whose terms are products two

---

[12]This is a very general constraint that we will also impose in AdS, but note that it is not rooted in entanglement structure. Only minima of the action dominate the semiclassical gravitational path integral.

[13]Another approach taken by [25] to studying DGP is to shift the tension by some amount proportional to the Ricci curvature of the branes. This cancels the explicit contribution of Ricci terms to the brane stress tensors and yields the equations (6).

first-derivative factors. Interestingly, [35] finds that the consistency of the bulk variational principle implies $a_i K_i \tilde{g}_i^{\mu\nu} + b_i K_i^{\mu\nu} = 0$, and contracting this against $K_{i,\mu\nu}$ reveals that the corresponding action vanishes on-shell. So, these terms do not alter the embedding equations (23). However, it is not immediate if or how these terms modify the entropy functional; see Section 4 for details. For simplicity, we only add a DGP term.

It is convenient to define a dimensionless parameter explicitly describing the relative strength of the DGP term against the RS term. We do so in terms of each brane's "effective" gravitational coupling, which is calculated in detail in Appendix A by integrating out the bulk extra dimension and isolating the $d$-dimensional Einstein–Hilbert term in the resulting effective action [25].[14] From that analysis, this coupling (the "effective Newton constant") on an brane with no DGP term is just

$$\frac{1}{G_{\text{RS}}} = \frac{L}{(d-2)G_{d+1}}. \tag{25}$$

If we apply that same procedure with a DGP term with coupling $G_{\text{b}}$ present, however, then we get

$$\left.\frac{1}{G_{\text{eff}}}\right|_{\text{RS+DGP}} = \frac{L}{(d-2)G_{d+1}} + \frac{1}{G_{\text{b}}} = \frac{1}{G_{\text{RS}}}\left(1 + \frac{G_{\text{RS}}}{G_{\text{b}}}\right). \tag{26}$$

So, in (22) we define the dimensionless parameters

$$\lambda_1 = \frac{G_{\text{RS}}}{G_{\text{b},1}}, \quad \lambda_2 = \frac{G_{\text{RS}}}{G_{\text{b},2}}, \tag{27}$$

as stand-ins for the couplings. In terms of $\theta_i$ and $\lambda_i$, the tension of brane $i$ is

$$T_i = \frac{d-1}{8\pi G_{d+1}L}\left(\cos\theta_i - \frac{\lambda_i}{2}\sin^2\theta_i\right). \tag{28}$$

We can and will trade the physical parameters $(T_i, G_{\text{b},i})$ for $(\theta_i, \lambda_i)$ in our analysis. The latter will turn out to be mathematically more convenient.

The limit $\lambda_1, \lambda_2 \to 0$ recovers the RS action (4). Meanwhile taking either $\lambda < -1$ yields a negative effective Newton constant. This leads to the "wrong sign" in the Euclidean actions, in that classical saddles would be maxima rather than minima.[15] Such saddles are actually suppressed in the gravitational path integral and thus ill-defined as semiclassical configurations, and so the region

$$\{\lambda_1 < -1\} \cup \{\lambda_2 < -1\}, \tag{29}$$

is considered pathological for these branes.

While the bulk classical solutions to (22) are essentially the same as for without DGP terms, there is a key change in applying the RT prescription that we must address. In the entropy functional, we pick up contributions from the areas of the intersection points between $\gamma_{\mathcal{R}}$ and the branes, in accordance with the semiclassical island rule [25, 56]. Specifically, the entropy functional (derived by [25]) is

$$S[\mathcal{R}] = \min_{\gamma_{\mathcal{R}} \sim \mathcal{R}} \text{ext}\left(\frac{A[\gamma_{\mathcal{R}}]}{4G_{d+1}} + \frac{\tilde{A}[\gamma_{\mathcal{R}} \cap \mathcal{Q}_1]}{4G_{\text{b},1}} + \frac{\tilde{A}[\gamma_{\mathcal{R}} \cap \mathcal{Q}_2]}{4G_{\text{b},2}}\right)$$

$$= \frac{1}{4G_{d+1}} \min_{\gamma_{\mathcal{R}} \sim \mathcal{R}} \text{ext}\left[A[\gamma_{\mathcal{R}}] + \frac{L}{d-2}\left(\lambda_1 \tilde{A}[\gamma_{\mathcal{R}} \cap \mathcal{Q}_1] + \lambda_2 \tilde{A}[\gamma_{\mathcal{R}} \cap \mathcal{Q}_2]\right)\right], \tag{30}$$

---

[14]Note that [25] takes two copies of a bulk geometry with a brane glued together via Israel junction conditions, whereas we consider one bulk geometry with an "end-of-the-world" brane. This changes the KR brane's effective Newton constant by a factor of 2 and also alters the effective cosmological constant.

[15]As mentioned above, this constraint—that the brane's $G_{\text{eff}}$ coupling must be positive—applies to flat-space DGP constructions, such as the original one of Dvali, Gabadadze, and Porrati [24].

where $A$ is a $(d-1)$-dimensional "bulk" area functional while $\tilde{A}$ is a $(d-2)$-dimensional "boundary" area functional. In the RS limit $\lambda_1, \lambda_2 \to 0$, we recover (9). The inclusion of boundary terms in the extremization does not alter the bulk equation for extremal curves $\gamma_{\mathcal{R}}$, but it does affect boundary conditions.

We reiterate that we apply both the positivity of horizon entropy and causal wedge inclusion (CWI) as swampland criteria on entanglement entropy. With that in mind, our goal is to use the DGP-modified entropy functional in the black-string solution (18) (Figure 4) to constrain the space of DGP couplings $(\lambda_1, \lambda_2)$. Technically the two-KR-brane RS + DGP models have four free parameters which include not just these two couplings but also the brane angles $(\theta_1, \theta_2)$. Thus we find $\theta$-dependent constraints on $\lambda_1$ and $\lambda_2$. This will allow us to explore how these constraints depend on the brane angles.

We should mention that our approach of using entanglement here builds on some previous work of [25], which finds DGP couplings yielding negative $G_{\text{eff}}$ (i.e. $\lambda < -1$) to be unphysical through using the entropy functional rather than the wrong-sign-action argument. Using a class of extremal surfaces in AdS described as "bubbles," [25] argues that the entropy functional supplemented by DGP terms (30) has no global minimum when $G_{\text{eff}} < 0$. Thus, the $\lambda_1, \lambda_2 < -1$ regimes of the DGP couplings are deemed pathological. Notably, the constraints we find are independent from the considerations of [25].

## 3.2 Extremal curves in the RS + DGP black string

We again consider the $(d+1)$-dimensional black string in the bulk with planar branes given by (18)–(19). For convenience, we fix $L = 1$. To reiterate, the metric and branes are

$$ds^2 = \frac{1}{u^2 \sin^2 \mu}\left[-h(u)dt^2 + \frac{du^2}{h(u)} + u^2 d\mu^2 + d\vec{x}^2\right], \quad h(u) = 1 - \frac{u^{d-1}}{u_{\text{h}}^{d-1}}, \tag{31}$$

$$\mathcal{Q}_1: \ \mu = \theta_1, \qquad \mathcal{Q}_2: \ \mu = \pi - \theta_2. \tag{32}$$

We take $\mathcal{R}$ to be the defect system at $t = 0$. The time-independent extremal curves may then be parameterized as

$$u = u(\mu), \quad \mu \in (\theta_1, \pi - \theta_2). \tag{33}$$

From the metric (31), we have that the bulk area functional is

$$A[u(\mu)] = V_{d-2}\int_{\theta_1}^{\pi-\theta_2} \frac{d\mu}{[u(\mu)\sin\mu]^{d-1}}\sqrt{\frac{u'(\mu)^2}{h[u(\mu)]} + u(\mu)^2}$$

$$\equiv V_{d-2}\int_{\theta_1}^{\pi-\theta_2} d\mu\, \mathcal{L}_s[\mu, u(\mu), u'(\mu)], \tag{34}$$

where $V_{d-2} = \int d\vec{x}$ and we denote the integrand as a Lagrangian $\mathcal{L}_s$. Furthermore, the boundary area functional for the intersection between $u(\mu)$ and $\mu = \mu_0$ is

$$\tilde{A}[u(\mu_0)] = \frac{V_{d-2}}{[u(\mu_0)\sin\mu_0]^{d-2}}. \tag{35}$$

And so, the entropy functional is

$$S[\mathcal{R}] = \frac{V_{d-2}}{4G_{d+1}}\min_{\gamma_{\mathcal{R}}\sim\mathcal{R}}\text{ext}\left(\int_{\theta_1}^{\pi-\theta_2} d\mu\, \mathcal{L}_s[\mu, u(\mu), u'(\mu)] + \frac{1}{d-2}\sum_{i=1}^{2}\frac{\lambda_i}{(u_i\sin\theta_i)^{d-2}}\right), \tag{36}$$

where we have defined $u_1 \equiv u(\theta_1)$ and $u_2 \equiv u(\pi - \theta_2)$ to make the notation more compact. Note that the DGP couplings $\lambda_1$ and $\lambda_2$ only explicitly appear in the boundary terms, although they will implicitly affect the extremal surfaces through boundary conditions.

Now, we extremize the functional in (36). Taking the variational derivative and integrating by parts yields (up to the fixed factor of $\frac{V_{d-2}}{4G_{d+1}}$)

$$\delta S \sim \int_{\theta_1}^{\pi-\theta_2} d\mu \left[ \frac{\partial \mathcal{L}_s}{\partial u} - \frac{d}{d\mu}\left(\frac{\partial \mathcal{L}_s}{\partial u'}\right) \right]\delta u - \left[ \left.\frac{\partial \mathcal{L}_s}{\partial u'}\right|_{\mu=\theta_1} + \frac{\lambda_1 \sin\theta_1}{(u_1 \sin\theta_1)^{d-1}} \right]\delta u_1$$
$$+ \left[ \left.\frac{\partial \mathcal{L}_s}{\partial u'}\right|_{\mu=\pi-\theta_2} - \frac{\lambda_2 \sin\theta_2}{(u_2 \sin\theta_2)^{d-1}} \right]\delta u_2 \,, \tag{37}$$

where $\delta u_1$ and $\delta u_2$ are first-order variations of the surface's endpoints. Setting the first term to 0 yields the Euler–Lagrange equation for $\mathcal{L}_s$, which we solve to write a second-order ODE for $u(\mu)$:

$$u'' = -(d-2)u\,h(u) + (d-1)u' \cot\mu \left[ 1 - \frac{\tan\mu}{2h(u)}\frac{u'}{u} + \frac{1}{h(u)}\frac{u'^2}{u^2} \right] - \left(\frac{d-5}{2}\right)\frac{u'^2}{u} \,. \tag{38}$$

The second and third terms are the boundary conditions on $\mathcal{Q}_1$ and $\mathcal{Q}_2$, respectively. Taking the dynamical boundary conditions [i.e. $\delta u_1 \neq 0$ and $\delta u_2 \neq 0$ in (37)] yields

$$\frac{u_1'}{\lambda_1} = -(\sin\theta_1)h(u_1)\sqrt{\frac{u_1'^2}{h(u_1)} + u_1^2} \leq 0\,, \quad \frac{u_2'}{\lambda_2} = (\sin\theta_2)h(u_2)\sqrt{\frac{u_2'^2}{h(u_2)} + u_2^2} \geq 0\,, \tag{39}$$

where we have defined $u_1' \equiv u'(\theta_1)$ and $u_2' \equiv u'(\pi-\theta_2)$. The inequalities follow from

$$\theta_i \in (0,\pi) \implies \sin\theta_i > 0\,, \quad u_i \in (0, u_h) \implies h(u_i) > 0\,, \quad u_i' \in \mathbb{R} \quad (i=1,2)\,, \tag{40}$$

and fix the signs of $\frac{u_1'}{\lambda_1}$ and $\frac{u_2'}{\lambda_2}$. With this in mind, we rearrange these expressions to write

$$u_1'^2 = \frac{(\lambda_1 \sin\theta_1)^2 h(u_1)^2 u_1^2}{1 - (\lambda_1 \sin\theta_1)^2 h(u_1)}\,, \quad u_2'^2 = \frac{(\lambda_2 \sin\theta_2)^2 h(u_2)^2 u_2^2}{1 - (\lambda_2 \sin\theta_2)^2 h(u_2)}\,, \tag{41}$$

and solve for $u_1'$ and $u_2'$:

$$u_1' = -\frac{(\lambda_1 \sin\theta_1)h(u_1)u_1}{\sqrt{1 - (\lambda_1 \sin\theta_1)^2 h(u_1)}}\,, \quad u_2' = \frac{(\lambda_2 \sin\theta_2)h(u_2)u_2}{\sqrt{1 - (\lambda_2 \sin\theta_2)^2 h(u_2)}}\,. \tag{42}$$

We immediately observe that the horizon $u(\mu) = u_h$ is still a solution even with the DGP boundary conditions above. This means that the horizon is always a valid candidate for the RT surface. We will use this fact in the following discussion.

## 3.3 Analytic criteria for DGP couplings

We now take an analytic approach to our exploration of the parameter space of DGP couplings $(\lambda_1, \lambda_2)$ by employing the RT prescription. By noting that the horizon of the black string is always at least a candidate surface, we can make the following assertions:

(1) Theories in which the RT prescription selects the horizon and computes a positive entropy cannot be excluded.

(2) The horizon entropy upper bounds the RT entropy.

In this section, we will use these statements to explore the DGP parameter space without numerically constructing extremal surfaces. The results are summarized in Figure 5. Furthermore, we reiterate that the region $\{\lambda_1 < -1\} \cup \{\lambda_2 < -1\}$ is already deemed unphysical by both our wrong-sign-action argument and an entropy argument by [25].

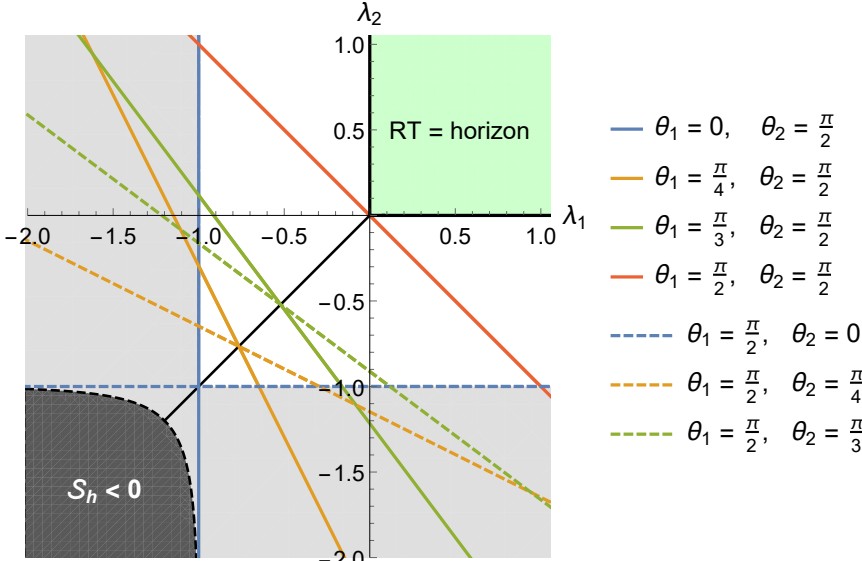

Figure 5: A plot of the two-brane DGP parameter space $(\lambda_1, \lambda_2)$ for $d = 4$. The upper-right (green) region $\{\lambda_1 \geq 0\} \cap \{\lambda_2 \geq 0\}$ cannot be excluded by our swampland criteria because the horizon is always the (positive-entropy) RT surface. The colored lines demarcate the boundaries of regions excluded by the positive-horizon-entropy requirement for fixed $\theta_1$ and $\theta_2$, which parameterize the branes as per (31)–(32). We exclude points below these lines. Additionally, the constraints have a reflection symmetry seen by swapping the labeling of the branes ($1 \longleftrightarrow 2$). We see that thinner wedges yield more restrictive bounds on $(\lambda_1, \lambda_2)$. The intersection of all $(\theta_1, \theta_2)$-dependent regions describes the couplings for which the horizon entropy is negative for all brane angles. This is shown in black and lies entirely within the unphysical $G_{\text{eff}} < 0$ region, with latter being shaded grey.

**Entanglement does not constrain $\{\lambda_1 \geq 0\} \cap \{\lambda_2 \geq 0\}$**   We first consider the case of non-negative DGP couplings on both branes. In this case, we find that our swampland criteria do not apply. Specifically, the entropy functional (36) is always positive, and we can and will also show that the RT surface is always the horizon, which is consistent with CWI. To do the latter, we study locally smooth extremal surfaces assumed to not be the horizon, finding that they are always UV-divergent and, thus, always subdominant to the (UV-finite) horizon in the functional integral that computes the entropy. In other words, we find that non-horizon candidates (i.e. finite extremal surfaces) for the RT prescription require either DGP coupling being negative.

Suppose that we have an extremal surface that is anchored to brane 1 at a point $u = u_1 < u_{\text{h}}$ and brane 2 at a point $u = u_2 < u_{\text{h}}$. Then, from the boundary conditions (42), we have that

$$u_1' < 0, \quad u_2' > 0. \tag{43}$$

Pictorially in Figure 4, such an extremal surface bends "away" from the horizon at both branes. Thus, $u'(\mu)$ must change sign at some $\mu = \mu_0$ somewhere in the bulk, i.e. there exists $\mu_0 \in (\theta_1, \pi - \theta_2)$ for which

$$u'(\mu_0 - \epsilon) < 0, \quad u'(\mu_0 + \epsilon) > 0, \tag{44}$$

for all sufficiently small $\epsilon > 0$. Since the metric is non-singular, the solutions to the Euler–Lagrange equation do not accommodate finite discontinuities in $u'$. We also observe that $u'(\mu_0)$

cannot vanish; if it does, then the equations of motion (38) give

$$u''(\mu_0) = -(d-2)u(\mu_0)h[u(\mu_0)], \tag{45}$$

where $u(\mu_0) \in (0, u_1)$ and thus $h[u(\mu_0)] > 0$. It then follows that $u''(\mu_0) < 0$, but this is inconsistent with (44).

The only remaining possibility is that $u'$ has an infinite discontinuity at $\mu = \mu_0$, i.e. that taking $\epsilon \to 0$ in (44) respectively yield $-\infty$ and $+\infty$. The resulting surface would not be globally smooth, but it would consist of two locally smooth components, where each reaches the defect at $u = 0$. Nonetheless, we can discard such a surface because it has a UV divergence in its entropy and thus is larger than the (UV-finite) horizon entropy.

We emphasize that the argument for this statement goes through if either DGP coupling is zero. Indeed, we had already proven the statement for $\lambda_1 = \lambda_2 = 0$ in [60]. For just one positive and one zero DGP coupling, we can set $\lambda_1 = 0$ without loss of generality. In this case, we must exploit the fact that $u''(\theta_1) = 0$ by the equation of motion (38). Thus, for some small $\epsilon \ll 1$, $u'(\theta_1 + \epsilon) < 0$ and, because we still have $u'_2 > 0$, there exists a $\mu = \mu_0$ where $u'(\mu)$ changes sign. However, we reiterate that such a sign change cannot happen without an infinite discontinuity. This completes the proof.

To summarize, for nonnegative DGP couplings $\lambda_1, \lambda_2 \geq 0$, the RT surface is always the horizon because the only other extant extremal surfaces are UV-divergent, so we never find a violation of CWI in this part of parameter space. Our search for CWI-violating theories in the swampland must involve at least one negative DGP coupling.

**Positivity of entropy constrains $\{\lambda_1 < 0\} \cup \{\lambda_2 < 0\}$**   For a $(d+1)$-dimensional AdS black string with two branes, the entropy *density*, which we define as the functional (36) in units of the prefactor $\frac{V_{d-2}}{4G_{d+1}}$, is computed to be

$$\mathcal{S}_{\rm h} = \frac{1}{u_{\rm h}^{d-2}} \left[ \int_{\theta_1}^{\pi - \theta_2} \frac{d\mu}{\sin^{d-1}\mu} + \frac{1}{d-2} \left( \frac{\lambda_1}{\sin^{d-2}\theta_1} + \frac{\lambda_2}{\sin^{d-2}\theta_2} \right) \right]. \tag{46}$$

It is evident that if either DGP coupling is sufficiently negative, then the horizon entropy is negative. While this is mathematically possible, we argue that this is unphysical.

We briefly reiterate the main points regarding horizon-entropy positivity discussed in Section 2. Recall that the RT prescription computes a von Neumann entropy, and such an entropy must be positive. In a typical AdS calculation with no branes, applying the RT prescription to a CFT subregion gives a UV-divergent result, since the surface reaches the conformal boundary. In the field theory, the divergence comes about because we have split the degrees of freedom. We often subtract this infinity via some ultraviolet renormalization procedure. The renormalized entropy can be negative (depending on the scheme), but this does not violate the original positivity requirement.

Meanwhile in a two-brane setup, the RT prescription gives UV-finite answers because we never split a connected CFT region. Another way to say this is that the algebra of observables on a defect system is type I, rather than type III. However, such an entropy computed from the gravity side may concernedly negative, which does not happen when there are no branes.

With that said, recall that the black-string horizon is extremal for all DGP couplings, since it solves (38) and (41). Thus, the horizon entropy upper-bounds the entropy $S$ from RT and

$$\mathcal{S}_{\rm h} < 0 \implies S \leq \frac{V_{d-2}}{4G_{d+1}} \mathcal{S}_{\rm h} < 0. \tag{47}$$

In other words, a negative horizon entropy implies a negative RT entropy.

Now we rule out parts of $(\lambda_1, \lambda_2)$ parameter space in which either DGP coupling is negative enough to induce a negative horizon entropy. First, consider fixed brane angles $\theta_1$ and $\theta_2$. The excluded region is then determined by a linear constraint:

$$
\begin{aligned}
\mathcal{S}_h &= \frac{1}{u_h^{d-2}} \left[ \int_{\theta_1}^{\pi-\theta_2} \frac{d\mu}{\sin^{d-1}\mu} + \frac{1}{d-2} \left( \frac{\lambda_1}{\sin^{d-2}\theta_1} + \frac{\lambda_2}{\sin^{d-2}\theta_2} \right) \right] < 0 \\
&\iff \frac{\lambda_1}{\sin^{d-2}\theta_1} + \frac{\lambda_2}{\sin^{d-2}\theta_2} < -(d-2) \int_{\theta_1}^{\pi-\theta_2} \frac{d\mu}{\sin^{d-1}\mu} .
\end{aligned}
\tag{48}
$$

The "bounding line" of this region depends on the values of $\theta_1$ and $\theta_2$. Furthermore, this constraint depends on the dimension parameter $d$. As such, for concreteness we set $d = 4$, discussing general $d$ in Appendix B. For $d = 4$, the horizon's entropy density is

$$
\begin{aligned}
\mathcal{S}_h &= \int_{\theta_1}^{\pi-\theta_2} \frac{d\mu}{u_h^2 \sin^3\mu} + \frac{\lambda_1}{2(u_h \sin\theta_1)^2} + \frac{\lambda_2}{2(u_h \sin\theta_2)^2} \\
&= \frac{1}{2u_h^2} \sum_{i=1}^{2} \left[ \csc\theta_i \cot\theta_i - \log\left( \tan\frac{\theta_i}{2} \right) + \lambda_i \csc^2\theta_i \right] .
\end{aligned}
\tag{49}
$$

We can then write the unphysical region as

$$
\lambda_2 < -\left( \frac{\sin\theta_2}{\sin\theta_1} \right)^2 \lambda_1 - \sin^2\theta_2 \sum_{i=1}^{2} \left[ \csc\theta_i \cot\theta_i - \log\left( \tan\frac{\theta_i}{2} \right) \right] .
\tag{50}
$$

Let us now explore more $\theta$-independent bounds. For convenience, we define notation for the fixed-$(\theta_1, \theta_2)$ unphysical regions in $(\lambda_1, \lambda_2)$ space and the corresponding bounding lines:

$$
\mathcal{B}(\theta_1, \theta_2) \equiv \{ \lambda_2 < L_\mathcal{B}(\lambda_1, \theta_1, \theta_2) \} ,
\tag{51}
$$

$$
L_\mathcal{B}(\lambda_1; \theta_1, \theta_2) \equiv -\left( \frac{\sin\theta_2}{\sin\theta_1} \right)^2 \lambda_1 - \sin^2\theta_2 \sum_{i=1}^{2} \left[ \csc\theta_i \cot\theta_i - \log\left( \tan\frac{\theta_i}{2} \right) \right] .
\tag{52}
$$

We can construct two $\theta$-independent regions: the *intersection* $\bigcap \mathcal{B}$ and the *union* $\bigcup \mathcal{B}$ of $\mathcal{B}(\theta_1, \theta_2)$ over all valid choices of brane angles. We do so in that order.

The intersection describes the part of parameter space in which the horizon entropy is always negative regardless of the brane angles. Thus, the intersection in the swampland. To find this region, it is helpful to define the contribution of an individual brane to the entropy:

$$
\mathcal{S}_h^{(i)} \equiv \frac{1}{2} \left[ \csc\theta_i \cot\theta_i - \log\left( \tan\frac{\theta_i}{2} \right) + \lambda_i \csc^2\theta_i \right] .
\tag{53}
$$

$\mathcal{S}_h^{(i)}$ has a positive singularity as $\theta_i \to 0^+$ for $\lambda_i \geq -1$. So, if either DGP coupling is at or above $-1$, then we can get a positive horizon entropy by taking the corresponding angle to be sufficiently small. This means that

$$
\bigcap \mathcal{B} \subset \{ \lambda_1 < -1 \} \cap \{ \lambda_2 < -1 \} .
\tag{54}
$$

Meanwhile, for $\lambda_i < -1$ the sign of the singularity flips, and $\mathcal{S}_h^{(i)}$ also has a negative singularity as $\theta_i \to \pi^-$. Thus, the profile of $\mathcal{S}_h^{(i)}$ develops a maximum in $\theta_i$ at

$$
\cos\theta_{i*} = -\frac{1}{\lambda_i} .
\tag{55}
$$

We can explicitly compute the value of this maximum:

$$\mathcal{S}_{i*} \equiv \mathcal{S}_{\mathrm{h}}^{(i)}\Big|_{\theta_i = \theta_{i*}} = \frac{\lambda_i}{2} - \frac{1}{4}\log\left(\frac{\lambda_i + 1}{\lambda_i - 1}\right). \tag{56}$$

Up to a positive factor, the total horizon entropy at a point in $\{\lambda_1 < -1\} \cup \{\lambda_2 < -1\}$ is upper-bounded by the sum of these maxima, and so

$$\mathcal{S}_{1*} + \mathcal{S}_{2*} < 0 \iff \mathcal{S}_{\mathrm{h}} < 0, \quad \forall\, (\theta_1, \theta_2). \tag{57}$$

Hence, the couplings $(\lambda_1, \lambda_2)$ for which $\mathcal{S}_{1*} + \mathcal{S}_{2*} < 0$ are precisely those in $\bigcap \mathcal{B}$. We find the boundary of this region by setting the sum of maxima to 0, which is equivalent to

$$\frac{\lambda_1 + \lambda_2}{2} - \frac{1}{4}\log\left[\left(\frac{\lambda_1 + 1}{\lambda_1 - 1}\right)\left(\frac{\lambda_2 + 1}{\lambda_2 - 1}\right)\right] = 0. \tag{58}$$

This produces a curve in $(\lambda_1, \lambda_2)$ space bounding the region of couplings for which horizon entropy is always negative, and thus we write

$$\bigcap \mathcal{B} = \left\{\frac{\lambda_1 + \lambda_2}{2} - \frac{1}{4}\log\left[\left(\frac{\lambda_1 + 1}{\lambda_1 - 1}\right)\left(\frac{\lambda_2 + 1}{\lambda_2 - 1}\right)\right] < 0\right\}. \tag{59}$$

The takeaway is that combinations of two *very* negative DGP couplings should be seen as be highly pathological, particularly since this regime is also ruled out by other arguments (namely our wrong-sign-action argument and the RT bubbles of [25]).

Next, the union only tells us where at least one set of brane angles furnishes a negative horizon entropy. Put another way, the complement of the union consists of the DGP parameters that are unaffected by the positivity of horizon entropy, and knowing the union helps visualize which part of the parameter space might conceivably be subjected to swampland constraints. With that in mind, we now compute the union. We argue that[16]

$$\bigcup \mathcal{B} = \{\lambda_1 < -1\} \cup \{\lambda_2 < -1\} \cup \{\lambda_1 + \lambda_2 < 0\}. \tag{60}$$

Notably, (60) contains positive-negative combinations of DGP couplings, including ones which yield $G_{\mathrm{eff}} > 0$ on both branes.

To prove (60), we first check that $\{\lambda_1 < -1\} \cup \{\lambda_2 < -1\} \cup \{\lambda_1 + \lambda_2 < 0\} \subset \bigcup \mathcal{B}$. To do so, we compute (51) in the following three limits:

$$\mathcal{B}\left(\frac{\pi}{2}, \frac{\pi}{2}\right) = \{\lambda_1 + \lambda_2 < 0\}, \tag{61}$$

$$\mathcal{B}\left(0, \frac{\pi}{2}\right) = \{\lambda_1 < -1\}, \tag{62}$$

$$\mathcal{B}\left(\frac{\pi}{2}, 0\right) = \{\lambda_2 < -1\}. \tag{63}$$

Now, we want to show that points that are not in the union of (61)–(63) also never accommodate negative horizon entropies. Since all of the fixed-$(\theta_1, \theta_2)$ bounds are linear in DGP couplings, we just need to confirm three statements for the bounding lines $L_{\mathcal{B}}$ (52):

(i) Any bounding line must have negative slope;

(ii) any bounding line must intersect $\lambda_1 = 0$ at some $\lambda_2 < 0$; and,

(iii) any bounding line which intersects $\lambda_1 + \lambda_2 = 0$ once does so in $\{\lambda_1 < -1\} \cup \{\lambda_2 < -1\}$.

---

[16]This turns out to be the union of all fixed-$(\theta_1, \theta_2)$ constraints in any number of spacetime dimensions. In contrast, the intersection depends on $d$. We show this in Appendix B.

These conditions ensure that any point contained in some excluded region $\mathcal{B}(\theta_1, \theta_2)$ will be in the union of of (61)–(63), thus proving (60). (i) is immediate from (52):

$$\frac{\partial L_{\mathcal{B}}}{\partial \lambda_1} = -\left(\frac{\sin\theta_2}{\sin\theta_1}\right)^2 < 0, \tag{64}$$

To check the other two statements, it helps to fix $\theta_2 = \theta$ and set $\theta_1 = \pi - \theta - \delta$, where $\delta \in (0, \pi - \theta)$. We then write

$$L_{\mathcal{B}}(0, \pi - \theta - \delta, \theta) = -\sin^2\theta \, F(\theta, \delta), \tag{65}$$

$$F(\theta, \delta) \equiv \csc\theta \cot\theta - \csc(\theta + \delta)\cot(\theta + \delta) - \log\left[\tan\left(\frac{\theta}{2}\right)\cot\left(\frac{\theta + \delta}{2}\right)\right]. \tag{66}$$

The derivative of $F$ with respect to $\delta$ is $2\csc^3(\theta + \delta)$, which is positive for all valid brane angles. Furthermore, $F(\theta, 0) = 0$, so we infer that $F > 0$. Thus, (65) is negative, confirming (ii).

To check (iii) directly,[17] we start by denoting the intersection between a bounding line $L_{\mathcal{B}}$ and $\lambda_1 + \lambda_2 = 0$ by $(\lambda_1^*, \lambda_2^*)$. We then use this intersection constraint to solve for $\lambda_1^*$ as a function of $\theta$ and $\delta$:

$$-\lambda_1^* = L_{\mathcal{B}}(\lambda_1^*, \pi - \theta - \delta, \theta) \implies \lambda_1^*(\theta, \delta) = \frac{F(\theta, \delta)}{\csc^2\theta - \csc^2(\theta + \delta)}. \tag{67}$$

Over the full domain of $\delta$, this function has the endpoint values

$$\lambda_1^*(\theta, 0) = \sec\theta, \quad \lambda_1^*(\theta, \pi - \theta) = -1. \tag{68}$$

We also observe that the derivative of $\lambda_1^*$ with respect to $\delta$ is positive for $\theta \in (0, \pi)$ and $\delta \in (0, \pi - \theta)$. So now if we assume $\theta > \frac{\pi}{2}$, then we have enough to deduce that $\lambda_1^* < -1$. However, if $\theta < \frac{\pi}{2}$, then we note that there is a pole at $\delta = \pi - 2\theta$. Nonetheless, the derivative is still positive, so it turns out that

$$\delta \in (0, \pi - 2\theta) \implies \lambda_1^* > 1, \quad \delta \in (\pi - 2\theta, \pi - \theta) \implies \lambda_1^* < -1. \tag{69}$$

The points for which $\lambda_1^* < -1$ are in $\{\lambda_1 < -1\}$. Meanwhile, since $\lambda_2^* = -\lambda_1^*$, the points for which $\lambda_1^* > 1$ are in $\{\lambda_2 < -1\}$. Thus, (iii) is true.

With (i)–(iii) in hand, we can conclude that any bounding line $L_{\mathcal{B}}(\lambda_1; \theta_1, \theta_2)$ must be contained within the union of (61)–(63), and so (60) is true. We reiterate that $\bigcup\mathcal{B}$ includes DGP parameters for which the effective Newton constant on each brane is positive, unlike $\bigcap\mathcal{B}$.

## 3.4 Numerical violations of causal wedge inclusion

From the equation of motion (38) and boundary conditions (42), we can numerically solve for the (static) extremal surfaces for particular choices of brane angles $(\theta_1, \theta_2)$ and DGP parameters $(\lambda_1, \lambda_2)$. We may then check whether or not there exists an extremal surface with total entropy strictly smaller than the horizon. This would be a violation of causal wedge inclusion (CWI) and put that combination of parameters in the swampland.

We first obtain a plethora of numerical extremal surfaces. Specifically, we shoot from brane 1 with fixed angle $\theta_1$ and over a range of initial parameters $u_1$ and couplings $\lambda_1$. Note that the value of $u_1'$ is set by (42). We then filter the list of these solutions down to those consistent with a particular value of $\theta_2$ at the second brane. For these remaining points, we compute the entropy density $\mathcal{S}_{\text{ext}}$ numerically and the entropy density of the horizon $\mathcal{S}_{\text{h}}$ analytically (49).

---

[17]We use a more indirect method in Appendix B which can be applied in any number of dimensions.

Then, we repeat the above process by shooting from brane 2. In the end, we have a large list of points $(\lambda_1, \lambda_2)$ on a fixed-$(\theta_1, \theta_2)$ plane for which there is a non-horizon extremal surface, along with the associated entropies.

With that list in hand, we delete any points which seem pathological, such as those with large (and thus potentially numerically unstable) areas. Guided by both the positivity of horizon entropy $\mathcal{S}_h > 0$ and possible cases (16)–(17) diagnosing whether or not we have a violation of CWI, we sort the points $(\lambda_1, \lambda_2)$ in this list into three subsets:

$$(\lambda_1, \lambda_2), \text{ for which } \mathcal{S}_{\text{ext}} > \mathcal{S}_h > 0, \tag{70}$$

$$(\lambda_1, \lambda_2), \text{ for which } \mathcal{S}_h < 0, \tag{71}$$

$$(\lambda_1, \lambda_2), \text{ for which } \mathcal{S}_h > \mathcal{S}_{\text{ext}} \text{ and } \mathcal{S}_h > 0. \tag{72}$$

(70) consists of points that cannot be ruled out by the entropic swampland conditions considered in this paper, but we emphasize that these points may still appear in the pathological region with at least one negative effective Newton constant $\{\lambda_1 < -1\} \cup \{\lambda_2 < -1\}$. Meanwhile, the regions (71) and (72) are unphysical on the fixed-$(\theta_1, \theta_2)$ plane. Specifically, the points in (71) must be contained within the analytically excluded region $\mathcal{B}(\theta_1, \theta_2)$ defined in (51). Points in (72) are not in the analytically excluded region since they furnish positive horizon entropy, but they still violate CWI.[18]

Note that a pair of DGP couplings in satisfying (71) or (72) for some combination of brane angles need not always be in the swampland. In fact, a pair $(\lambda_1, \lambda_2)$ is only generically be in the swampland (based on our criteria) if it is in the union of (71) with (72) and taken over *all* $(\theta_1, \theta_2)$. The excluded points presented below (Figure 6) should be interpreted with a specific combination of brane angles in mind.

The primary fact that want to demonstrate is that this last set of points overlaps with the part of the $(\lambda_1, \lambda_2)$ parameter space that is not already excluded by either the positivity of horizon entropy or the positivity of the brane's effective Newton constant, i.e. that (72) includes points for which [recalling (51)–(52)]

$$\lambda_2 > L_{\mathcal{B}}(\lambda_1; \theta_2, \theta_2), \text{ and } \lambda_1, \lambda_2 > -1. \tag{73}$$

We can be more ambitious and ask for a complete understanding of the DGP couplings that allow for non-horizon extremal surfaces, as well as how such a set partitions into subsets (70)–(72). While our approach does this to some extent, we emphasize that our numerics provide only evidence for such information, rather than proof. However, the numerics are still sufficient to exclude a region of parameter space within (73).

**Inspecting the found points**  For various combinations of fixed $(\theta_1, \theta_2)$, we present the resulting sets of points in Figure 6. We observe points in (70) (red), (71) (blue), and (72) (green). By definition, the blue points are below the bounding line (52) along which $\mathcal{S}_h = 0$, while the red and green points are above this bounding line. Based solely on these numerics, we make several observations.

First, note that the blue points are in the region $\{\lambda_1 < -1\} \cap \{\lambda_2 < -1\}$, i.e. they appear at couplings for which both effective Newton constants are negative. This suggests that any couplings which allow negative horizon entropy and a non-horizon extremal surface are deeply pathological, since they violate multiple swampland criteria. Furthermore, these points are completely disconnected from the others.

The red and green points appear to form a single connected region above the bounding line $L_{\mathcal{B}}$ (52). Most of the red points, which we reiterate correspond to couplings that do

---

[18]*A priori* this set may also include more strongly pathological points for which $\mathcal{S}_{\text{ext}} < 0$. Not only would such points violate CWI, but they would also have a negative entropy as determined by the RT prescription.

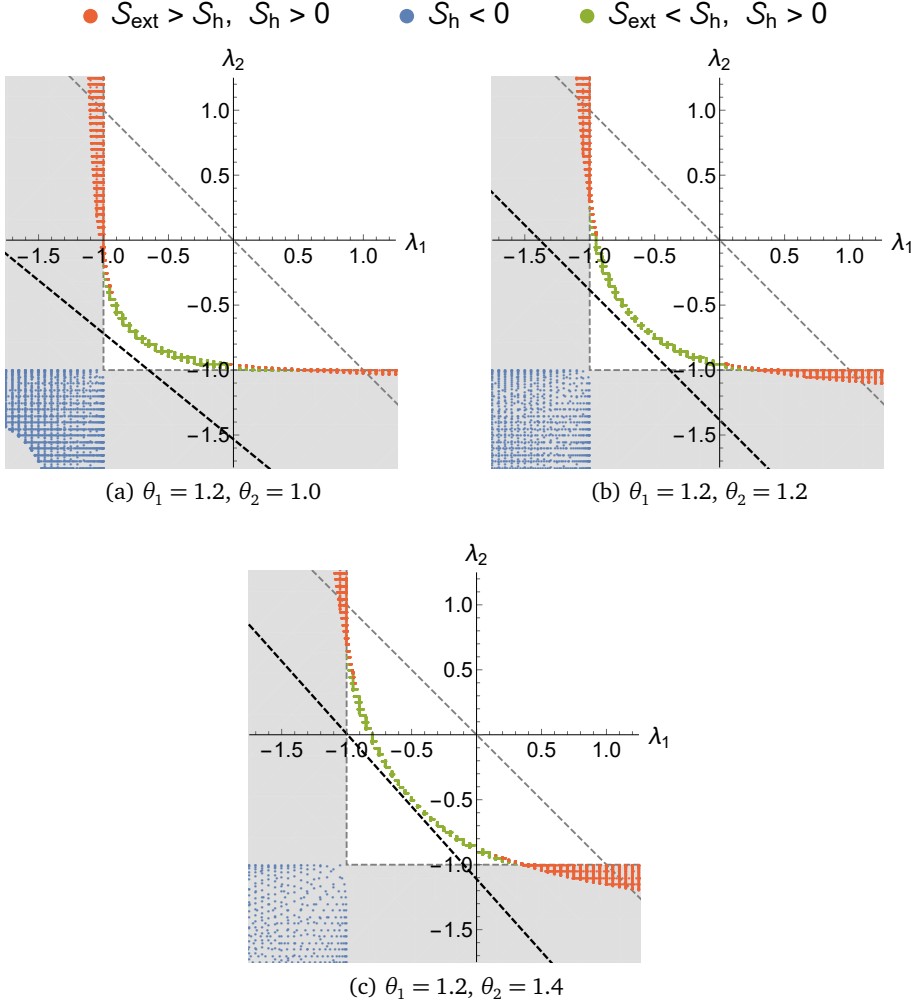

(a) $\theta_1 = 1.2$, $\theta_2 = 1.0$

(b) $\theta_1 = 1.2$, $\theta_2 = 1.2$

(c) $\theta_1 = 1.2$, $\theta_2 = 1.4$

Figure 6: The points in $(\lambda_1, \lambda_2)$ parameter space for which we find non-horizon extremal surfaces, for fixed angles $\theta_1$ and $\theta_2$. The colors correspond to (70) (red), (71) (blue), and (72) (green). The thick dashed lines are the $(\theta_1, \theta_2)$-dependent bounding lines (52) below which $\mathcal{S}_{\mathrm{h}} < 0$, while the light dashed line is $\lambda_1 + \lambda_2 = 0$. We have also shaded the region in which either effective Newton constant is negative.

not violate CWI, are pathological anyway because they furnish at least one negative effective Newton constant, i.e. $\lambda_1 < -1$ or $\lambda_2 < -1$. Interestingly, some are not sick, and these points are near but notably not arbitrarily close to the $\{\lambda_1 < -1\} \cup \{\lambda_2 < -1\}$ region. However, we caution that this part of the parameter space near the boundary of $\{\lambda_1 < -1\} \cup \{\lambda_2 < -1\}$ is subject to higher numerical errors and such points may be artifacts of those errors. We will elaborate on this point later.

Meanwhile, the green points designating CWI-violating couplings appear within the region $\{\lambda_1 > -1\} \cap \{\lambda_2 > -1\} \cap \{\lambda_1 + \lambda_2 < 0\}$.[19] So for a given $\theta_1$ and $\theta_2$, CWI excludes points not already ruled out by the positivity of horizon entropy and effective Newton constant. Thus, CWI provides another nontrivial swampland criteria for braneworlds.

We should say that these numerics do not suggest that CWI is a *stronger* condition than the positivity of horizon entropy. This would only be a true statement if the set of points ruled out by CWI contains the region $\mathcal{B}(\theta_1, \theta_2)$ (51) in which $\mathcal{S}_{\mathrm{h}} < 0$, but our numerics only find non-horizon extremal surfaces within a subset of $\mathcal{B}(\theta_1, \theta_2)$.

---

[19]Some green points are hidden below the red points, but even those green points are within this region.

How does this story depend on the combination of angles? While we have sampled and shown only a few combinations as shown in Figure 6, we can make some conjectures based on the observed patterns. For one thing, the green and red points are above the bounding line $L_{\mathcal{B}}$ but seem to follow the line as the angles are dialed. Numerically the CWI-excluded points are above the bounding line by a gap of order 0.1 at most in the sampled angles in Figure 6. For combinations of brane angles that yield a "wider" wedge, i.e. smaller $\theta_1 + \theta_2$, this line resides within the region $\{\lambda_1 < -1\} \cup \{\lambda_2 < -1\}$, with the limit $\theta_1, \theta_2 \to 0$ completely assuring positivity of entropy for all $(\lambda_1, \lambda_2)$. So if the CWI-excluded points must move with the bounding line, then in the limit $\theta_1, \theta_2 \to 0$ we would expect CWI to become redundant with the positivity of the effective Newton constants.

It would be worthwhile to search for CWI-violating surfaces in the regime of small brane angles. However, taking both $\theta_1, \theta_2 \to 0$ gives rise to instabilities in our numerics, so we are unable to confidently rule out points in this regime of parameter space. A more refined approach to searching for the CWI-violating surfaces is necessary. Alternatively it may be possible to organize the contributions of higher-derivative terms to the entropy into an ordered series in this limit, which would allow for analytic statements.

For "thinner" wedges, the bounding line intersects the region with positive effective Newton constants, i.e. increasing $\theta_1 + \theta_2$ also pushes $L_{\mathcal{B}}$ up and to the right in the $(\lambda_1, \lambda_2)$ parameter space. This also pushes the red and green points up, and such combinations of brane angles is when CWI distinguishes itself from the other swampland conditions.

**Possible expansion of excluded zone** The key takeaway is that CWI certainly excludes DGP couplings that otherwise seem okay. The green points in Figure 6 are certainly sick. However, there are various nontrivial features of the plots. In particular, the switch between the green region and red region seems somewhat arbitrary, and there is also a gap between the red/green region and the bounding line $L_{\mathcal{B}}$. This latter feature in particular suggests that CWI does not actually lower-bound the DGP couplings.

To be more precise, on the fixed-$(\theta_1, \theta_2)$ slices shown in Figure 6, the excluded zone is not convex and consists of two disconnected pieces. But we reiterate that the disconnectedness is not expected to be a feature of the excluded manifold in the full four-dimensional parameter space. This is because of our expectation that in the $\theta_1, \theta_2 \to 0$ limit, both CWI and positivity of entropy become subsumed by the condition that the effective Newton constants are positive, which by itself excludes a connected region.

We caution that the numerics could be subject to some slight instability originating from the shooting procedure, particularly when dealing with surfaces which are numerically close to the horizon. Such surfaces appear in our numerics near the boundary of the $\{\lambda_1 < -1\} \cup \{\lambda_2 < -1\}$ region. Furthermore, in our approach we only completely specify one of the DGP couplings, whereas we read off the other coupling from a list of solutions. Thus, some of the green or red points are only really specified up to some error, and if the difference between $\mathcal{S}_{\mathrm{h}}$ (computed analytically) and $\mathcal{S}_{\mathrm{ext}}$ (computed numerically) is within the margin of error of the shooting then the switch between red points and green points is also ambiguous. However, we emphasize that our numerics are relatively stable against changes in precision, and so we believe these errors to be insignificant for most couplings—particularly for green points that are further from the red region in Figure 6.

It should be possible to further improve on our numerics. We emphasize that our primary goal was to find points excluded only by CWI. We have accomplished that much by considering a range of angles in which the numerics are under control. However, if one wants to better understand how the excluded zone is embedded in parameter space [for example, the shape of the excluded points in $(\lambda_1, \lambda_2, \theta_1)$ keeping $\theta_2$ fixed], then it would be necessary to use more refined approaches.

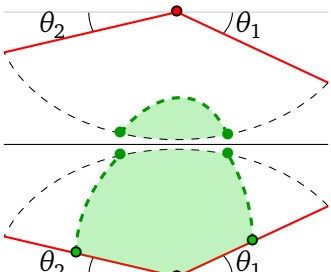

Figure 7: A cartoon of a possible time-dependent entangling surface in the two-brane setup with DGP couplings, shown on the $t = 0$ slice of the maximally extended black-string geometry, in which the horizons are identified. This surface would violate causal wedge inclusion—we can see that the entanglement wedge (in green) does not include the full exterior of either side and thus never includes the causal wedge of either defect.

We also do not search for potential extremal surfaces that go into the black-string interior (Figure 7) and thus correspond to a time dependence in the entanglement entropy for the one-sided defect. Such surfaces require that $\lambda_1 < 0$ and $\lambda_2 < 0$ due to their shape, but they are harder to construct because they are not $\mathbb{Z}_2$ symmetric with respect to the maximally extended geometry.[20] Nonetheless, these would also violate causal wedge inclusion and could in principle be used to rule out theories, possibly filling the gap between the CWI-excluded and positive-entropy-excluded regions observed in Figure 6.

However, it is possible that such surfaces are not even mathematically feasible. In AdS black geometries without branes, entanglement surfaces akin to Figure 7 and anchored to the conformal boundary are not allowed because horizons in those settings are "extremal surface barriers" [86]. So, the only way to get time-dependent surfaces without branes is to compute the entropy of a boundary region supported on both sides of the maximally extended geometry (cf. [87]). But, it is not clear that such a mathematical obstruction exists for surfaces ending on KR branes. If it does, then it would be a mathematical statement—rather than just a physical one—that the entanglement entropy of the one-sided defect geometry can only follow a trivial (time-independent) Page curve.

## 4 Discussion

To summarize, we have explored how entanglement structure constrains effective field theories of gravity on branes in AdS. Our core philosophy is that the AdS/CFT dictionary, being a feature of UV physics, gives rise to swampland criteria that may be implemented in semiclassical holographic models. As a case study, we have used this idea to put nontrivial constraints—both analytic and numerical—on the RS + DGP braneworld model of gravity.

We should emphasize that the end-of-the-world branes we consider in this paper are of a particular class. They are "constant-angle" branes (32) embedded in a bulk AdS space (31). In principle, one may consider other foliations of the bulk into KR branes. This would yield more configurations, and in principle holography could be used to put different constraints on higher-derivative brane couplings. We leave this to future explorations.

Another notable caveat is that we only consider one particular two-derivative correction

---

[20]One may consider surfaces that are anchored to the branes, dive into the horizon, and satisfy this $\mathbb{Z}_2$ symmetry as in [67]. However, those surfaces contribute to the entropy functional of the *purified* defect system. This entropy is 0, and the corresponding bulk RT surface "shrinks" onto the brane's horizon [38].

to the brane action—the DGP term. Even at the level of two derivatives in the effective theory, we can write additional independent terms proportional to squares of extrinsic curvature. Schematically, this correction takes the form

$$I_{K^2} = \int_{\text{brane}} (aK^2 + bK^{\mu\nu}K_{\mu\nu}). \tag{74}$$

So in truth, general two-brane setups with actions truncated at two derivatives occupy an eight-dimensional parameter space parameterized as $(\theta_1, \theta_2, \lambda_1, \lambda_2, a_1, a_2, b_1, b_2)$. Fortunately as found by [35], (74) does not affect the embedding equations (23), so the shapes of the branes are only determined by the RS tensions and DGP couplings. However, (74) may alter the $(d-2)$-dimensional area terms in the entropy functional (30). To see how, one would need to apply the recipe of [25] [specifically their equation (A.7)] to (74).

Our goal in this paper was merely to show the possibility that a swath of effective brane-localized theories can be excluded on general physical principles. We do this by restricting to a slice of theory space where the terms (74) (and all higher couplings) are dialed to zero. We leave exploration of other higher-derivative couplings to future work.

**Trivial massless Page curves**   We briefly comment on the significance of these results to the recent discourse on the black hole information paradox. The typical $(d > 2)$ higher-dimensional realization of Page curves as describing the evolution of semiclassical black hole information relies on the construction with one-KR-brane embedded in an ambient AdS geometry with a nontrivial blackening factor. The first such construction was numerical [14], and a simple analytical model was provided by [57] shortly after. Moreover, [57] claimed that the existence of a nontrivial Page curve may be due to the semiclassical theory on the brane being a *massive* gravity theory, a feature that has long been ascribed to the presence of a non-gravitating bath glued on at asymptotic infinity [58,59].

This aspect of the one-brane setup motivated our study of a two-brane setup embedded in a bulk black string geometry in [60]. Here, one of the branes acts as a gravitating bath, and so the semiclassical theory on the branes maintains a massless graviton in its spectrum [61]. We had found the entropy of radiation collected in the gravitating bath to be eternally computed by the bulk horizon area. Thus, the entropy of radiation was found to be constant in time, in agreement with the claims of [64].

Recent work [67–69] has proposed that our prior result is changed by adding DGP terms to the two-brane setup. The basic claim is that for at least some combinations of DGP couplings (with at least one being negative) the bulk horizon no longer computes the minimum of the entropy functional, allowing for time-dependent RT surfaces. Furthermore, one may also get static RT surfaces that are not the horizon. For example in making this case, [67] explicitly finds the latter type of surface for a particular combination of parameters, which in our conventions are[21]

$$\theta_1 \approx 1.090, \quad \theta_2 \approx 0.585, \quad \lambda_1 = 0, \quad \lambda_2 \approx -0.984. \tag{75}$$

However, the DGP couplings allowing for *any* such non-horizon RT surfaces in the first place are in conflict with causal wedge inclusion, since the causal wedge of one of the defect systems includes the entire corresponding side's exterior region. That is also true of time-dependent surfaces (see Figure 7).[22] Thus, semiclassical massless-gravity theories furnishing nontrivial

---

[21]As a consistency check, we have checked our entropies associated with these values against those of [67]. The analytic horizon entropies are precisely the same. Numerically, shooting from either brane yields $\mathcal{S}_{\text{ext}} \approx 0.838$, which is consistent with the value $\mathcal{S}_{\text{ext}} \approx 0.842$ of [67] up to small $O(0.1\%)$ error.

[22]As a side note, explicitly constructing time-dependent surfaces may allow one to further expand the excluded zone presented in Figure 6. There may also exist brane angles for which time-dependent surfaces are the *only* way to violate CWI, but searching for such points is beyond the methods in the current paper.

time-dependent Page curves or any static non-horizon bulk RT surfaces are in the swampland, and the lessons of [67] are unphysical. See [38] for more specific discussion of this point.

In fact, our numerics loosely indicates that most (but conceivably not all) DGP couplings for which there even *exists* a UV-finite extremal surface apart from the horizon are sick, either because of a violation of causal wedge inclusion or some other physical pathology such as a negative effective Newton constant.

In conclusion, we have shown how swampland constraints can restrict the viability of AdS gravity theories with a holographic interpretation. Indeed, holography is fundamentally a feature of UV physics, and so care must be applied in assuming that it holds semiclassically to avoid misleading conclusions. Our work here focuses on entanglement structure in braneworld models, with an eye towards their recent application to the black hole information problem. The takeaway is that the existence of a holographic dictionary is an important limitation on physically consistent semiclassical models of gravity.

## Acknowledgments

We are grateful to Suvrat Raju for collaboration during the initial stages of this project and for reviewing the manuscript. We also thank Elena Cáceres for useful discussions and Dominik Neuenfeld for comments on a previous version of this manuscript.

**Funding information** The work of HG is supported by the grant (272268) from the Moore Foundation "Fundamental Physics from Astronomy and Cosmology." AK, MR, and MY are supported in part by the U.S. Department of Energy under Grant No. DE-SC0022021 and a grant from the Simons Foundation (Grant 651440, AK). CP is supported by the NSF Grant No. PHY-2210562 and the Robert N. Little Fellowship. The work of LR is supported by NSF Grant Nos. PHY-1620806 and PHY-1915071, the Chau Foundation HS Chau postdoc award, the Kavli Foundation grant "Kavli Dream Team," and the Moore Foundation Award 8342. SS is supported by NSF Grant Nos. PHY-2112725 and PHY-2210562.

## A  Effective action with RS + DGP gravity

We briefly review the derivation of the semiclassical gravitational action on the brane with both an RS and a DGP term. We discuss this in the case of a semi-infinite $(d+1)$-dimensional bulk geometry $\mathcal{M}$ with one $d$-dimensional brane $\mathcal{Q}$ (as in [25, 85, 88, 89]), but the basic results carry over to the two-brane configuration. The starting point is the bulk action with brane-localized terms (leaving Gibbons–Hawking terms implicit),

$$I = \underbrace{\frac{1}{16\pi G_{d+1}} \int_{\mathcal{M}} \sqrt{-g}\left[R + \frac{d(d-1)}{L^2}\right]}_{\text{Einstein–Hilbert}} + \underbrace{\int_{\mathcal{Q}} \sqrt{-\tilde{g}}\left(-T + \frac{1}{16\pi G_{\text{b}}}\tilde{R}\right)}_{\text{RS + DGP}}, \qquad \text{(A.1)}$$

where $g$ is the bulk metric, $\tilde{g}$ is the induced brane metric, and tildes denote curvature invariants of the latter. We implicitly work with a bulk metric of the form

$$ds^2 = \frac{L^2}{u^2 \sin^2 \mu}\left[-h(u)dt^2 + \frac{du^2}{h(u)} + d\vec{x}^2 + u^2 d\mu^2\right], \qquad \text{(A.2)}$$

where $u > 0$, $(t, \vec{x}) \in \mathbb{R}^{d-1}$, and $\mu \in (\theta, \pi)$. $\mu = \theta$ designates the position of the brane, and $\theta$ is geometrically the angle between this brane and the conformal boundary. $h(u)$ is some analytic function for which this metric satisfies Einstein's equations (5).

We then treat the brane like a cutoff surface and integrate the Einstein–Hilbert term over the bulk radial direction. [25] computes the resulting dimensionally-reduced action to be of the form

$$I_{\text{diver}} = \frac{1}{16\pi G_{d+1}} \int \sqrt{-\tilde{g}} \left[ \frac{2(d-1)}{L} + \frac{L}{d-2} \tilde{R} + O(\tilde{R}^2) \right]. \tag{A.3}$$

There is an infinite tower of higher-derivative corrections, but in the semiclassical regime we are simply concerned with the Einstein–Hilbert terms on the brane consisting of $\tilde{R}^0$ and $\tilde{R}^1$ terms. Because we are dealing with just one copy of the bulk instead of two (cf. [25]), the effective action includes only one copy of $I_{\text{diver}}$:

$$I_{\text{eff}} = I_{\text{diver}} + \int_{\mathcal{Q}} \sqrt{-\tilde{g}} \left( -T + \frac{1}{16\pi G_b} \tilde{R} \right). \tag{A.4}$$

Plugging in (A.3) then yields

$$I_{\text{eff}} = \int_{\mathcal{Q}} \sqrt{-\tilde{g}} \left[ \left( \frac{d-1}{8\pi G_{d+1} L} - T \right) \tilde{R}^0 + \left( \frac{L}{16\pi(d-2)G_{d+1}} + \frac{1}{16\pi G_b} \right) \tilde{R} + O(\tilde{R}^2) \right]. \tag{A.5}$$

We then write the action in the form

$$I_{\text{eff}} = \frac{1}{16\pi G_{\text{eff}}} \int_{\mathcal{Q}} \sqrt{-\tilde{g}} \left[ \frac{(d-1)(d-2)}{\ell_{\text{eff}}^2} + \tilde{R} + O(\tilde{R}^2) \right], \tag{A.6}$$

where the $G_{\text{eff}}$ is the effective Newton constant (i.e. the coupling of the effective Einstein–Hilbert terms) and $\ell_{\text{eff}}$ is the effective curvature scale,[23] each of which are respectively identified as

$$\frac{1}{G_{\text{eff}}} = \frac{L}{(d-2)G_{d+1}} + \frac{1}{G_b}, \qquad \frac{1}{\ell_{\text{eff}}^2} = \frac{2}{L^2} \left( 1 - \frac{8\pi G_{d+1} L}{d-1} T \right) \left[ \frac{(d-2)G_{d+1}}{G_b L} + 1 \right]^{-1}. \tag{A.7}$$

## A.1 The $\lambda$ and $\theta$ parameters

Throughout this paper, we study a particular class of "planar" KR branes (32) with AdS$_d$ geometry embedded in a bulk AdS$_{d+1}$ spacetime (31). We find it more useful to parameterize the embeddings of our branes not by $(T, G_b)$ but by alternate parameters $(\theta, \lambda)$. $\theta$ measures the angle between the brane and the conformal boundary while $\lambda$ captures the relative contributions between the RS and DGP terms to the effective Newton constant. We now define the mapping between these different sets of parameters.

Taking $\frac{1}{G_b} \to 0$ in (A.1) to remove the DGP term, we are simply left with the RS term. The effective RS Newton constant and curvature scale are then identified as

$$\frac{1}{G_{\text{RS}}} = \frac{L}{(d-2)G_{d+1}}, \qquad \frac{1}{\ell_{\text{RS}}^2} = \frac{2}{L^2} \left( 1 - \frac{8\pi G_{d+1} L}{d-1} T \right). \tag{A.8}$$

The RS Newton constant partly contributes to the effective Newton constant of the brane with RS + DGP terms (A.7). This motivates us to define a dimensionless parameter,

$$\lambda \equiv \frac{G_{\text{RS}}}{G_b}, \tag{A.9}$$

---

[23]We emphasize that this effective curvature scale is not the same as the induced curvature radius of the brane read off the induced metric, e.g. (20).

describing the relative strength of these two terms. The effective Newton constant is then

$$\frac{1}{G_{\text{eff}}} = \frac{1}{G_{\text{RS}}}(1 + \lambda).$$ (A.10)

In particular, we have a negative effective Newton constant if $\lambda < -1$.

Next, we write the dynamical boundary condition of the brane with a DGP term. In RS, recall that it is (6). More generically, the boundary condition takes the form

$$K_{\mu\nu} - K\tilde{g}_{\mu\nu} = T_{\mu\nu},$$ (A.11)

where $T_{\mu\nu}$ is the brane's stress tensor (with a factor of $8\pi G_{d+1}$ included in the definition),

$$T_{\mu\nu} = -\frac{16\pi G_{d+1}}{\sqrt{-\tilde{g}}} \frac{\delta I_{\mathcal{Q}}}{\delta \tilde{g}^{\mu\nu}}.$$ (A.12)

For the brane action in (A.1), this is simply

$$\begin{aligned}
T_{\mu\nu} &= -8\pi G_{d+1} T \tilde{g}_{\mu\nu} - \frac{G_{d+1}}{G_{\text{b}}}\left(\tilde{R}_{\mu\nu} - \frac{1}{2}\tilde{R}\tilde{g}_{\mu\nu}\right) \\
&= -8\pi G_{d+1} T \tilde{g}_{\mu\nu} - \frac{\lambda L}{d-2}\left(\tilde{R}_{\mu\nu} - \frac{1}{2}\tilde{R}\tilde{g}_{\mu\nu}\right).
\end{aligned}$$ (A.13)

We have used the relationship between $G_{\text{RS}}$ and $G_{d+1}$ to write the right-hand side in terms of $\lambda$. We can then solve for the tension by contracting (A.11) with the induced metric tensor. This yields

$$8\pi G_{d+1} T = \frac{d-1}{d} K + \frac{\lambda L}{2d}\tilde{R}.$$ (A.14)

Now by computing the curvature invariants associated with $\mu = \theta$ in (A.2),

$$K = \frac{d}{L}\cos\theta, \quad \tilde{R} = -\frac{d(d-1)}{L^2}\sin^2\theta,$$ (A.15)

we can at last write the tension as a function of $\lambda$ and $\theta$:

$$T = \frac{d-1}{8\pi G_{d+1} L}\left(\cos\theta - \frac{\lambda}{2}\sin^2\theta\right).$$ (A.16)

Equipped with (A.9) and (A.16), we may use the parameters $(\theta, \lambda)$ to fix the explicit couplings $(T, G_{\text{b}})$. We thus parameterize the effective theories in terms of the former.

## A.2 Flat-space limits

The DGP term is often used to construct models of flat branes in flat space [24], with the RS tension being a bare vacuum energy [90]. However, our paper is about swampland criteria emerging from AdS/CFT, and so we are considering the DGP term in the context of AdS branes in AdS space as in [25]. Nonetheless, to connect to braneworld phenomenology, we can ask what happens to our constraints by taking one of two flat limits.

Roughly speaking, the first limit is to take the bulk curvature radius $L$ to be large (making the bulk cosmological constant vanish) and the brane tension to be 0. The second is a flat limit of just the effective theory on the brane in which we take the effective length scale (A.7) to be large. However, we find that the constraints from entanglement lose all power in either case.

Let us be more specific. The first flat-space limit we can consider is one in which we take

$$L \to \infty, \quad T \to 0, \quad G_{d+1}, G_{\text{b}}\text{-fixed}.$$ (A.17)

In other words, we are eliminating the cosmological-constant terms in (A.1). This yields the original DGP model [24] with flat branes in a flat bulk. Instead of (A.5), the resulting semi-classical theory on the brane is

$$I_{\text{eff}} \xrightarrow{L \to \infty, \ T \to 0} \frac{1}{16\pi G_{\text{b}}} \int \sqrt{-\tilde{g}} \left[ \tilde{R} + O(\tilde{R}^2) \right],$$ (A.18)

i.e. we have that $G_{\text{eff}} = G_{\text{b}}$ and $\ell_{\text{eff}} \to \infty$. Note that in this regime, the requirement that $G_{\text{eff}} > 0$ turns into the constraint $G_{\text{b}} > 0$.

(A.17) is perfectly compatible with the semiclassical regime of the Ryu–Takayanagi prescription $G_{d+1} \ll L^{d-1}$. However, by keeping $G_{d+1}$ and $G_{\text{b}}$ finite while sending $L \to \infty$, we are essentially imposing the limit

$$\lambda = \frac{G_{\text{RS}}}{G_{\text{b}}} = \frac{(d-2)G_{d+1}}{L G_{\text{b}}} \to 0.$$ (A.19)

Our entanglement-based swampland criteria, which require at least one negative DGP coupling in the two-brane setup, are no longer applicable.

Another possibility is to consider a "flat-brane" limit in which the effective cosmological constant $\sim \frac{1}{\ell_{\text{eff}}^2}$ vanishes. This can be done by taking the following limit of the tension:

$$T \to \frac{d-1}{8\pi G_{d+1} L}.$$ (A.20)

For a given $\lambda$, we can rephrase this in terms of a limit on the brane angle $\theta$. From (A.16), we get (A.20) by taking one of two values for the brane angle:

$$\theta \to 0^+, \ \text{ or } \ \theta \to \text{Cos}^{-1}\left(1 + \frac{2}{\lambda}\right).$$ (A.21)

For the first brane angle, we are essentially taking the planar brane to the conformal boundary. In the case of two planar branes forming a wedge, this makes the wedge wider. As seen in the main text however, the excluded points would then be pushed into the $\{\lambda_1 < -1\} \cup \{\lambda_2 < -1\}$ region, in which $G_{\text{eff}} < 0$ on at least one of the branes. Meanwhile, the second brane angle only makes sense when we assume $\lambda < -1$. So for this flat limit, we see that the swampland constraints coming from entanglement are simply subsumed by the requirement that $G_{\text{eff}} > 0$ and thus lose all power.

However, we should emphasize that we are working with a particular slicing whereby the brane is a constant-$\mu$ hypersurface in (A.2). It is conceivable that another configuration of branes could yield well-defined bounds in flat-space limits. We leave this to future work.

## B  Dimension dependence of the positive-entropy constraint

In general, the entropy density of the $(d+1)$-dimensional black string's horizon with two RS + DGP branes is given by (46), which we recall here:

$$\mathcal{S}_{\text{h}} = \frac{1}{u_{\text{h}}^{d-2}} \left[ \int_{\theta_1}^{\pi-\theta_2} \frac{d\mu}{\sin^{d-1}\mu} + \frac{1}{d-2} \left( \frac{\lambda_1}{\sin^{d-2}\theta_1} + \frac{\lambda_2}{\sin^{d-2}\theta_2} \right) \right].$$ (B.1)

$\theta_1$ and $\theta_2$ are the angles of the branes while $\lambda_1$ and $\lambda_2$ are their respective DGP couplings. Furthermore, $d$ is the number of brane dimensions. Physically, (B.1) must never be negative,

but mathematically there exist ranges of $\lambda_1$ and $\lambda_2$ for which $\mathcal{S}_{\text{h}} < 0$. For fixed brane angles $(\theta_1, \theta_2)$, we denote this part of the $(\lambda_1, \lambda_2)$ parameter space by

$$\mathcal{B}_d(\theta_1, \theta_2) \equiv \left\{ \int_{\theta_1}^{\pi - \theta_2} \frac{d\mu}{\sin^{d-1}\mu} + \frac{1}{d-2}\left( \frac{\lambda_1}{\sin^{d-2}\theta_1} + \frac{\lambda_2}{\sin^{d-2}\theta_2} \right) < 0 \right\}, \tag{B.2}$$

and assert that it resides in the swampland.

In the main text (Section 3.3), we use this condition to rule out a section of the DGP parameter space for $d = 4$. For fixed angles, the constrained region is linear, and we also compute both the *intersection* $\bigcap \mathcal{B}_d$ and the *union* $\bigcup \mathcal{B}_d$ of these excluded regions over all choice of $(\theta_1, \theta_2)$. The intersection is the region of DGP parameter space in which we get a negative horizon entropy for any choice of brane angles, while the union is that in which we find a negative horizon entropy for at least one choice of brane angles. However, the details may depend on $d$. Our goal in this appendix is to describe this dependence. Roughly speaking, the intersection becomes smaller as $d$ is tuned larger, but the union is $d$-independent.

**Intersection of excluded regions**     First, we observe that the horizon-entropy density can be rewritten as a sum over the two branes:

$$\mathcal{S}_{\text{h}} = \frac{1}{u_{\text{h}}^{d-2}} \sum_{i=1}^{2} \mathcal{S}_{\text{h}}^{(i)}(d), \qquad \mathcal{S}_{\text{h}}^{(i)}(d) \equiv \int_{\theta_i}^{\frac{\pi}{2}} \frac{d\mu}{\sin^{d-1}\mu} + \frac{\lambda_i}{d-2}\csc^{d-2}\theta_i. \tag{B.3}$$

As in the $d = 4$ case, $\mathcal{S}_{\text{h}}^{(i)}$ generally[24] diverges as $\theta_i \to 0^+$ or $\theta_i \to \pi^-$. The signs of these divergences depend on $\lambda_i$, and the divergence at $\theta_i = 0$ is particularly important for determining the intersection. If there is a positive divergence there, then we may force the entire sum to be positive by taking $\theta_i \ll 1$, .

We can find the signs of these divergences by examining the derivative of $S_{\text{h}}^{(i)}$ with respect to $\theta_i$,

$$\frac{\partial \mathcal{S}_{\text{h}}^{(i)}}{\partial \theta_i} = -\csc^{d-1}\theta_i\,(1 + \lambda_i \cos\theta_i). \tag{B.4}$$

For $|\lambda_i| < 1$, this quantity is negative, and so $S_{\text{h}}^{(i)}$ monotonically decreases. Thus the divergence as $\theta_i \to 0^+$ must be positive, while the divergence as $\theta_i \to \pi^-$ is negative. For $|\lambda_i| > 1$, however, the derivative has a zero at $\theta_i = \theta_{i*}$, where

$$\cos\theta_{i*} = -\frac{1}{\lambda_i}. \tag{B.5}$$

Furthermore, we observe that $\lambda_i > 1$ implies that this is a minimum, whereas $\lambda_i < -1$ implies it is a maximum. So, both divergences are positive for $\lambda_i > 1$, whereas they are negative for $\lambda_i < -1$.

Taken together, these observations imply that if either $\lambda_1 > -1$ or $\lambda_2 > -1$, then we may take one of the angles to be very small, inducing a positive blow-up in the total entropy. And so,

$$\bigcap \mathcal{B}_d \subset \{\lambda_1 < -1\} \cap \{\lambda_2 < -1\}. \tag{B.6}$$

We had found this in $d = 4$, but we have just shown that (B.6) is true in any number of dimensions. Recalling that $\{\lambda_1 < -1\} \cup \{\lambda_2 < -1\}$ is already deemed pathological both because of our wrong-sign-action argument in the main text and the RT bubble argument of [25], (B.6) means that the DGP couplings for which we get negative horizon entropy for all combinations of brane angles are already sick.

---

[24]The exception is for $d = 3$, for which the setting $\lambda_i = 1$ cancels the $\theta_i \to \pi^-$ divergence and taking $\lambda_i = -1$ cancels the $\theta_i \to 0^+$. These are edge cases, so we ignore them.

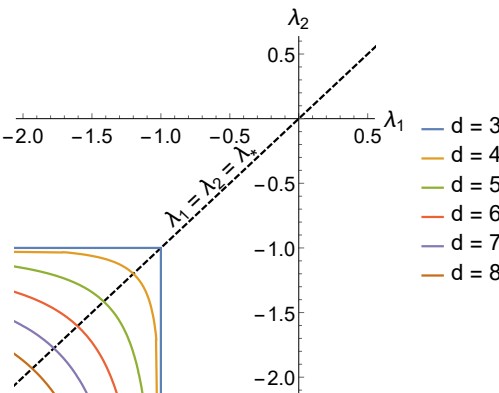

Figure 8: The $(\lambda_1, \lambda_2)$ parameter space with the $d$-dependent contours demarcating the intersection (over brane angles) of all regions excluded by horizon-entropy positivity. We depict contours for a variety of (brane) dimensions $d$. The intersection for a given $d$ is the region below the corresponding contour.

To compute the intersection more precisely, we must specify $d$. Upon doing so, we compute the maximum value of $\mathcal{S}_h^{(i)}(d)$ as a function of $\lambda_i$. For example, these maxima for the first three values $d = 3, 4, 5$ are

$$\mathcal{S}_h^{(i)}(3)\Big|_{\theta_i=\theta_{i*}} = -\sqrt{\lambda_i^2 - 1}, \tag{B.7}$$

$$\mathcal{S}_h^{(i)}(4)\Big|_{\theta_i=\theta_{i*}} = \frac{\lambda_i}{2} - \frac{1}{4}\log\left(\frac{\lambda_i + 1}{\lambda_i - 1}\right), \tag{B.8}$$

$$\mathcal{S}_h^{(i)}(5)\Big|_{\theta_i=\theta_{i*}} = \frac{1}{3}\frac{2 - \lambda_i^2}{\sqrt{\lambda_i^2 - 1}}. \tag{B.9}$$

The point is that a pair of couplings $(\lambda_1, \lambda_2)$ is in $\bigcap \mathcal{B}_d$ precisely when the sum of the maxima for brane 1 ($i = 1$) and brane 2 ($i = 2$) is negative. We can find this region by finding the contour in $(\lambda_1, \lambda_2)$ parameter space along which the sum of maxima are 0, then by noting that the horizon entropy becomes negative if either DGP coupling decreases. In other words, the intersection of excluded regions is everything to the lower-left of the zero contour. These contours for various dimensions are shown in Figure 8.

**Union of excluded regions** We can also ask what part of the $(\lambda_1, \lambda_2)$ parameter space allows for *at least* one combination of brane angles furnishing a negative entropy, so as to get a sense of how much of the parameter space could potentially be impacted by the swampland constraints.[25] This turns out to be

$$\bigcup \mathcal{B}_d = \{\lambda_1 < -1\} \cup \{\lambda_2 < -1\} \cup \{\lambda_1 + \lambda_2 < 0\}, \tag{B.10}$$

regardless of the number of dimensions. The argument is similar to that of the main text. First, we note that

$$\mathcal{S}_h^{(i)}(d) = \frac{\sqrt{\pi}\,\Gamma\left(1 - \frac{d}{2}\right)}{2\Gamma\left(\frac{3}{2} - \frac{d}{2}\right)} - \frac{1}{2}B\left(\sin^2\theta_i; 1 - \frac{d}{2}, \frac{1}{2}\right) + \frac{\lambda_i}{d - 2}\csc^{d-2}\theta_i, \tag{B.11}$$

---

[25] To be more specific, the complement of the union is the part of the parameter space in which we will never get a negative horizon entropy and thus have no theories ruled out by the positivity requirement.

where $B$ is the incomplete beta function and we employ a dimensional regularization scheme[26] to write a finite entropy. We then take the following extremal combinations of brane angles:

$$\theta_1 \to \frac{\pi}{2}, \quad \theta_2 \to \frac{\pi}{2} \implies \mathcal{S}_{\mathrm{h}}^{(1)}(d) + \mathcal{S}_{\mathrm{h}}^{(2)}(d) \sim \frac{1}{d-2}(\lambda_1 + \lambda_2), \tag{B.12}$$

$$\theta_1 \to 0, \quad \theta_2 \to \frac{\pi}{2} \implies \mathcal{S}_{\mathrm{h}}^{(1)}(d) + \mathcal{S}_{\mathrm{h}}^{(2)}(d) \sim \frac{(1+\lambda_1)}{d-2} \csc^{d-2}\theta_1, \tag{B.13}$$

$$\theta_1 \to \frac{\pi}{2}, \quad \theta_2 \to 0 \implies \mathcal{S}_{\mathrm{h}}^{(1)}(d) + \mathcal{S}_{\mathrm{h}}^{(2)}(d) \sim \frac{(1+\lambda_2)}{d-2} \csc^{d-2}\theta_1. \tag{B.14}$$

For these respective combinations of brane angles, requiring positive horizon entropy excludes the region $\{\lambda_1 < -1\}$, $\{\lambda_2 < -1\}$, and $\{\lambda_1 + \lambda_2 < 0\}$, and so we at least have

$$\bigcup \mathcal{B}_d \subset \{\lambda_1 < -1\} \cup \{\lambda_2 < -1\} \cup \{\lambda_1 + \lambda_2 < 0\}. \tag{B.15}$$

Now, we argue that any point outside of the set on the right-hand side cannot furnish negative horizon entropy for any combination of angles. To do so, we write the general-$d$ bounding line in slope-intercept form:

$$\lambda_2 = L_{\mathcal{B},d}(\lambda_1; \theta_1, \theta_2) \equiv -\left(\frac{\sin\theta_2}{\sin\theta_1}\right)^{d-2}\lambda_1 - (d-2)\sin^{d-2}\theta_2 \int_{\theta_1}^{\pi-\theta_2} \frac{d\mu}{\sin^{d-1}\mu}. \tag{B.16}$$

As the constraint for fixed $(\theta_1, \theta_2)$ is linear, it is sufficient to show the following three statements, which we had shown for $d = 4$ in the main text:

(i) Any bounding line must have negative slope;

(ii) any bounding line must intersect $\lambda_1 = 0$ at some $\lambda_2 < 0$; and,

(iii) any bounding line which intersects $\lambda_1 + \lambda_2 = 0$ once does so in $\{\lambda_1 < -1\} \cup \{\lambda_2 < -1\}$.

(i) is immediately true; the slope read from (B.16) is always negative. (ii) is true because the integral $\int \csc^{d-1}\mu$ is of a positive function over a positive interval ($\theta_1 < \pi - \theta_2$), and $\sin\theta_2$ is also positive.

(iii) is difficult to prove directly in general $d$, but we can prove it indirectly by using the linearity of the bound for fixed $(\theta_1, \theta_2)$ in conjunction with (i) and (ii). Denote the intersection of a bounding line $L_{\mathcal{B},d}$ with $\lambda_1 + \lambda_2 = 0$ by $(\lambda_1^*, \lambda_2^*)$. Then, for any $L_{\mathcal{B},d}$ with a negative slope and negative $\lambda_2$-intercept, $\lambda_1^* \in (-1, 0)$ implies that $L_{\mathcal{B},d}$ intersects $\lambda_1 = -1$ at some $\lambda_2 > 1$, whereas $\lambda_1^* \in (0, 1)$ implies that $L_{\mathcal{B},d}$ intersects $\lambda_2 = -1$ at some $\lambda_1 > 1$.

So, we now show that the bounding line evaluated at $\lambda_1 = -1$ must have a value below 1. This argument along with the symmetry of the problem under interchanging $1 \leftrightarrow 2$ is sufficient to also assert that bounding lines intersect $\lambda_2 = -1$ only at $\lambda_1 < 1$. Fixing $\theta_2 = \theta$ and setting $\theta_1 = \pi - \theta - \delta$ where $\delta \in (0, \pi - \theta)$, we evaluate

$$L_{\mathcal{B},d}(-1; \pi - \theta - \delta, \theta) = \frac{\sin^{d-2}\theta}{\sin^{d-2}(\theta+\delta)} - (d-2)\sin^{d-2}\theta \int_{\pi-\theta-\delta}^{\pi-\theta} \frac{d\mu}{\sin^{d-1}\mu}. \tag{B.17}$$

At $\delta = 0$, this evaluates to 1. Furthermore, its derivative is

$$\frac{\partial L_{\mathcal{B},d}}{\partial\delta}\bigg|_{\lambda_1 = -1} = -(d-2)\frac{\sin^{d-2}\theta}{\sin^{d-1}(\theta+\delta)}[1 + \cos(\theta+\delta)] < 0. \tag{B.18}$$

The punchline is that a bounding line $L_{\mathcal{B},d}$ cannot actually intersect $\lambda_1 = -1$ at a point where $\lambda_2 > 1$. By essentially the same reasoning, this line cannot intersect $\lambda_2 = -1$ at a point where

---

[26]The dimensional regulator cancels the even-$d$ divergence of the beta function, allowing us to ignore it.

$\lambda_1 > 1$. Lastly, we note that $\lambda_1^* \neq 0$ unless the bounding line is itself $\lambda_1 + \lambda_2 = 0$. Ergo, $\lambda_1^* \notin (-1, 1)$, and so (iii) is true.

With (i)–(iii) proven, we conclude that any excluded region $\mathcal{B}_d(\theta_1, \theta_2)$ is contained within the union $\{\lambda_1 < -1\} \cup \{\lambda_2 < -1\} \cup \{\lambda_1 + \lambda_2 < 0\}$. Thus, (B.10) is true for any number of spacetime dimensions.

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
