# Peer review of "Constraining braneworlds with entanglement entropy"

_SciPost Physics, doi:SciPost Phys. 15, 199 (2023)_

## Round 1 · Referee Report · Anonymous (Referee 1) · 2023-8-31

Report

This paper studies certain consistency constraints on braneworld scenarios in AdS. The constraints come from two properties of the holographic entanglement entropy: it is positive and must obey causal wedge inclusion (CWI). The authors consider a system of two branes, anchored on a shared surface at the boundary of AdS, in the background of a black string. The action of the branes is taken to be the Randall-Sundrum one (\emph{i.e.} the tension term) augmented by an Einstein-Hilbert term, known as the Dvali-Gabadadze-Porrati (DGP) term. The authors show that the entropy constraints rule out certain regions in coupling space.

Presently, the main motivation of interest in these models is in relation to the black hole information paradox. Indeed, these branes support a massless graviton, and the entropy exchange between the two branes can be shown to follow the expected time-dependent Page curve, for appropriate choices of couplings. The most consequential claim of the authors is that such choices of couplings are in the swampland, thus strengthening the conclusion that non-trivial islands, which lead to the Page curve, only exist in theories with a massive graviton.

Both the approach and the conclusions of this work are interesting. The paper is well written and, as far as I can see, correct. I will be happy to recommend it for publication after the authors have addressed the following observations.

  1. In the paragraph before eq. 1.2, two sentences contain the unnecessary repetition of the same concept, with the same exact words: "[the branes are] coarse descriptions of warped compactifications in string theory".
  2. More importantly, it is not known---at least to me---what generality can be attributed to the statement itself. The available UV complete construction of boundary conditions for holographic CFTs often do not look like Karch-Randall branes at low energy, and might for instance involve extra dimensions shrinking smoothly instead. The authors do not claim differently, but it would be better to add a paragraph which points this out explicitly, since the reader might get the impression that the theories considered here are universal low energy effective field theories for CFTs with defects. Some literature in this context should be cited as well, including for instance 2108.10345, 2110.05491, 2212.14058 and some additional references to the appropriate string theory constructions.
  3. At the bottom of page 4, there is a typo: ''the Von Neumann entropy of the entropy of the defect''.
  4. In the outline---section 1.2---the authors summarize their finding that a ''range of combinations of DGP couplings are disallowed''. This allows me to state the only criticism about the exposition of the results. The region in coupling space which ends up in the swampland is never fully explained. It is a region in the four dimensional space spanned by the two tensions and the DGP couplings (or equivalently by $\theta_i$ and $\lambda_i$), so it cannot be drawn on paper. Part of this region is analytically known, though: the one excluded by the criterion of positivity of the entropy. The authors draw a projection on this region in figure 5: it would be useful to refer to this figure, and to the equations that determine the full region in $(\theta_i,\lambda_i)$, already in the introduction. Vice versa, the region which is put in the swampland by CWI is only explored numerically, and results are reported only for a few values of the brane tensions. Nevertheless, figure 6 also deserves a reference in the introduction, so that the reader can easily decipher the main results of the paper.
  5. In section 2.1, the modification of the RT prescription for the brane setup is explained. When the authors declare to use Neumann boundary conditions for the end-points of the RT surface on the brane, 1105.5165 (ref. 67) should be cited, since this prescription was proposed there. Furthermore, it would be useful to comment on its status. To the best of my knowledge, no complete proof à la Lewkowycz-Maldacena is available in this case. Vice versa, arguments in favor of its validity were presented in appendix A of 2006.04851 and in section 5.1 of 2202.11718.
  6. On page 10, the ideas behind CWI are outlined. In view of the importance of this fact for the results on the paper, it would be advisable to comment explicitly on the extension of the theorem of reference 36 to the case of wedge holography.
  7. On page 15, the authors discuss the possibility of adding other two derivative couplings beyond the DGP term. They disregard this possibility because of the unclear effect of these terms on the prescription for the computation of entanglement entropy. This is an important caveat, and should be mentioned in the introduction or in the conclusions. It also solicits a more general question about the robustness of the swampland constraints. Reliable EFT constraints are statement about low energy Wilson coefficients which are independent of all higher derivative couplings: they state, for instance, that no UV completion exists for certain values of these coefficients. Is this the case for the entropy constraints? The constraints only seem to rule out theories with precisely the considered Lagrangian, since further higher derivative modifications might change the entropy functional and the brane embeddings. Is there an argument why higher derivative couplings are expected to give parametrically small corrections to the bounds obtained here? In other words, is the entanglement entropy a quantity perturbatively computable in the derivative expansion? Either way, the authors should comment about this issue in the paper.
  8. On page 19, the proof of non-extremality for non-horizon surfaces in the case of positive DGP coupling on the branes is explained. I find the phrasing of this proof slightly inaccurate, despite the conclusion being correct. In eq. 3.23, the inequalities should not be strict. Having both limits to be zero (\emph{i.e.} a smooth minimum for the curve) is allowed. More to the point, this equation is not the relevant one to prove a contradiction with $u''(\mu_0)<0$. Indeed, the value of (the limit of) the first derivative of a function is not constraining for the value of the second derivative. What the authors probably meant is that in a left neighborhood of $\mu_0$ $u'(\mu_0)<0$ and in a right neighborhood $u'(\mu_0)>0$, which indeed is incompatible with $u''(\mu_0)<0$. Finally, the proof leaves open the logical possibility of a finite discontinuity in the derivative (rather than the infinite one considered in the paper). In this case, $u''(\mu_0)$ is not defined, while the left and right limits $u''(\mu_0^\pm)$ exist and might have any value in principle. This case should be excluded on general grounds: the Euler-Lagrange equations should have smooth solutions if the metric is non-singular. The authors might want to point this out explicitly.
  9. Above eq. 3.32, the authors write that ''it is natural to'' place points which disobey positivity of the entropy in the swampland. This kind of vague phrasing is used in multiple occasions throughout the paper. Why is this? Theories with precisely these values of the coupling (with the caveat of higher derivative corrections raised above) are rigorously excluded by the analysis, so the authors should make plain statements: these points are in the swampland. If, on the other hand, other caveats make the results of the paper non-rigorous, those concerns should be discussed.
  10. In section 3.4, the requirement of CWI is explored numerically. Figure 6 contains a few examples of points which are put in the swampland. Are these the only values of the tension for which the numerics was run? It would be interesting and more complete to have an idea of the shape of the excluded region as a function of $\theta_i$ as well. Could the authors for instance draw a $3d$ shape of the boundary of the green region in $(\lambda_1,\lambda_2,\theta_1)$ at fixed $\theta_2$?
  11. On a related note, it is important to emphasize that the values of $\lambda_i$ which obey 3.51 are \emph{not} automatically in the swampland. While implied in the paper, I feel that this is not stated clearly enough, for instance around eqs. 3.49-3.51. A point in the $\lambda_i$ plane is only excluded if it lies in the swampland for all values of the tensions. Besides being relevant to understand the boundaries of the swampland, this consideration has consequences on the main application of the paper's results, \emph{i.e.} the (non-)existence of a time dependent Page curve.
  12. At the bottom of page 24, it is said that the CWI-violating points lie in $\left{\lambda_1<-1\right} \cap \left{\lambda_1<-1\right} \cap \left{\lambda_1+\lambda_2<0\right}$. This looks like a typo: should it be $\left{\lambda_1>-1\right} \cap \left{\lambda_1>-1\right} \cap \left{\lambda_1+\lambda_2<0\right}$?
  13. On page 25, the paragraph which starts with ''How does this story depends on the combination of angles?'' contains rather vague statements, like excluded points being ''close'' to the bounding line and the CWI condition becoming ''weaker'' in a certain loosely defined region in parameter space, where the wedge is ''wider'' (the quotation marks are from the authors!). In accordance with a previous comment, a more quantitative characterization of the swampland would go a long way: for instance, the authors could say if in the limits $\theta_1,\,\theta_2 \to 0$ the swampland reduces to the basic region of negative effective Newton constants on the branes. This limit is especially interesting with regards to the previous comment on higher derivative corrections to the action. Indeed, it corresponds to weakly coupled gravity on the brane. In this regime, higher derivative terms are expected to give parametrically suppressed contributions to on-shell quantities, and the swampland criteria of this paper should become precise. The authors could also sharpen the loose discussion in this paragraph with a plot of, say, the intersection of the green points with the $\lambda_1=\lambda_2$ line as a function of $\theta_1=\theta_2$.
  14. On page 26, the authors point out that the CWI-violating couplings do not bound the swampland. Again, this would be an occasion to comment more explicitly on the topology of the swampland, as defined in this paper. It is not convex and its projection onto the fixed $\theta_i$ plane has two disconnected components. Considerations on the $\theta_1,\,\theta_2 \to 0$ limit should also clarify if it is a disconnected manifold also in the full four-dimensional space.
  15. In the conclusions, the unphysical nature of two-brane models giving rise to a non-trivial Page curve is affirmed. However, it is not clearly stated if \emph{all} of the parameter space which gives results like in ref. 60 is in the swampland. The authors again seem to avoid sharp statements (''we would assert'', ''we would put [these models in the swampland]''), but more importantly, two features of these models are mentioned but their relation is not fully clarified. On one hand, a time-dependent Page curve arises from time dependent RT surfaces. On the other hand, ''one may also get static RT surfaces which are not the horizon''. The latter feature automatically violates CWI and are in the swampland. But it is the former feature the important one for the black hole information paradox, and violation of CWI for time dependent surfaces is not explored in the present paper. Are there values of the couplings which allow for time dependent Page curves, but do not lie in the swampland as defined by the authors' results?
  16. A final question, which the authors could clarify in the paper---or just answer in their response to this report---pertains locality. The boundaries of the swampland show a non-trivial dependence of the couplings of one brane on those of the other. This is a natural outcome of the procedure, but is surprising on general grounds: the UV physics, which the swampland is supposed to uncover, should be local. Is this situation due to the branes touching on the boundary of AdS? Would the swampland criteria become local, if the branes were to be separated, thus defining the dual of a CFT on a slab?
  • validity: -
  • significance: -
  • originality: -
  • clarity: -
  • formatting: -
  • grammar: -

Author:  Sanjit Shashi  on 2023-09-26  [id 4008]

(in reply to Report 1 on 2023-08-31)

We thank Referee 1 for their comments. Here are our responses to each of their points.

  1. We have consolidated the iterated clause pointed out by the referee.

  2. In the first paragraph, we have added a footnote clarifying that by "brane-like object," we really mean a sharp localized sheet of energy with one spatial length smaller than the AdS length scale and whose dynamics can thus be described by a low-energy effective action read from its derivative expansion. We have also added a footnote on the first page expressing that not all holographic boundary conditions are described by brane-like objects. We have cited both the literature mentioned by the referee that explores this subtlety and a swath of the literature about the Janus class of supergravity solutions corresponding to a defect but without a thin end-of-the-world brane.

  3. We have fixed the typo.

  4. We have slightly reworded the relevant paragraph in the outline to make clear that we are working with the $(\theta,\lambda)$ parameters, rather than the $(T,G_{\text{b}})$ parameters defined in the actions. We have also added references to the figures.

  5. We have made the suggested clarifications on the status of holographic entanglement entropy in AdS/BCFT in a new footnote.

  6. We have added a new paragraph at the very end of Section 2.1 discussing the role of CWI in wedge holography. Our basic point is that one should only need the notion of a causal wedge and an entanglement wedge to use it. The former can be constructed through causal curves again, while the latter is built from the surface obtained through the modified RT prescription in the wedge.

  7. First regarding the broader question, the starting point would be to look at equation (A.7) in the first "Islands Made Easy" paper \href{https://arxiv.org/abs/2006.04851}{arXiv:2006.04851}, which shows the result of applying the replica trick to these defect geometries. All higher-derivative terms will only affect the coefficient of the $(d-2)$-dimensional term. Furthermore, the strength of a given term's contribution will depend on some dimensionless combination between its coupling and overall bulk scales. Indeed, this is what has been seen with the DGP term in the literature and is written mathematically in our equation (3.9). In other words, it does not seem that the strength of higher-derivative terms are automatically organized with respect to one another in the entropy functional. As stated by the referee, we are taking a rather conservative approach by ruling out theories with a specific value or range of couplings. In fact, we would stress that our focus on RS + DGP (excluding even the other two-derivative terms) is a choice that essentially means our story takes place on a slice of a much larger theory space. We now state this caveat explicitly in the Discussion.

  8. We have made the refinements to the proof suggested by the referee, both swapping out the limit notation for $\epsilon$-notation and noting that finite discontinuities in the first derivative are disallowed by Euler--Lagrange since the metric is non-singular.

  9. This was an issue with our phrasing. As the referee states, the points in the $(\lambda_1,\lambda_2,\theta_1,\theta_2)$ parameter space that violate entropy-positivity or CWI \textit{are} excluded. We have removed the uses of the word "natural" so as to be more direct. We have also done away with the vague wording surrounding our claims (e.g. phrases like "we would assert" or "we would put this into the swampland") throughout the document in favor of sharper phrasing (e.g. "this is in the swampland").

  10. It might be possible to do this with more refined numerics. We chose a corridor of angles in which the numerics appeared to be stable. Additionally as our only goal was to show that a cluster of points can indeed be excluded by CWI but not by positivity, more thoroughly characterizing the exclusion space was not a priority. We have added a new paragraph at the end of section 3, under the discussion about "possible expansion of excluded zone," clarifying caveat.

  11. Shortly after the equations referenced by the referee (all in the start of Section 3.4), we have made more explicit the fact that we are working on a fixed-$(\theta_1,\theta_2)$ plane. We have also added a new paragraph clearly emphasizing that the excluded points of this section should be thought of with a specific pair of brane angles in mind, and that a pair $(\lambda_1,\lambda_2)$ is only completely excluded if it lies in the union of (3.50) with (3.51) and taken over all brane angles.

  12. We have fixed the typo.

  13. Regarding this paragraph, to be clear the behavior discussed here is actually conjectural. Admittedly there were some additional numerics that we had run consistent with our assertions here [i.e. values of $\theta_1,\theta_2$ for which the CWI-excluded points were within $O(0.05)$ of $\lambda_2 = -1$], but we chose not to include those in the document because they were not as trustworthy. Indeed, the main reason we did not more systematically construct the $\theta_1 = \theta_2$ plots as $\theta_1,\theta_2 \to 0$ is because the numerics break down quite dramatically if we take either brane angle to be too small. We have edited the text to be more clear on this. We have also elaborated on the potential of the $\theta_1,\theta_2 \to 0$ limit if either the numerics can be refined or if one can explicitly organize the higher-derivative contributions to entanglement.

  14. We have added a short paragraph about the topology of the excluded region. We have also mentioned that, based on our conjecture about the $\theta_1,\theta_2 \to 0$ limit, we do not expect the full 4d excluded manifold to be disconnected (since both entropy-positivity and, conjecturally, CWI should be subsumed by the constraint $G_{\text{eff}} > 0$).

  15. We have again changed to wording to be more sharp in the referenced passage. Our intent was to say that any couplings that do not give the horizon as the RT surface in a two-brane setup are in contradiction with causal wedge inclusion. This includes time-dependent surfaces. To see this, note that a time-dependent surface anchored to the two branes will always exclude some piece of the exterior geometry on the $t = 0$ slice of the entanglement wedge. We had some discussion of this in and around Figure 7, but we have also added a clause stating this reasoning in the relevant part of the Discussion. To be clear, we did not explicitly search for couplings that might furnish time-dependent surfaces but not static non-horizon surfaces. Our specific analysis was not suited to searching for such surfaces. However, it is still a good question and so we have added a footnote stating the possibility of such surfaces and their potential to rule out more of the theory space.

  16. As the referee says, this apparent nonlocality natural from the procedure. To be more specific, it comes from the fact that the entropy functional has both of the couplings in it. The von Neumann entropy is a nonlocal quantity, so constraints on the holographic entropy computed through some effective gravitational action might also be nonlocal in that sense. So, while we are borrowing the philosophy of "swampland" (since the thing we are really relying on is the consistency of AdS/CFT, which can be said to come from the UV physics), our approach does appear to be different from more typical bounds studied in the swampland program. For the specific question posed by the referee about whether this apparent nonlocality would go away if the branes were separated, the answer is no, subject to some qualifiers contingent on which entropy is being calculated. If we are concerned with the entropy of the full slab (so we are not splitting any subregions), then the entropy would be computed by surfaces anchored to both branes and thus pick up contributions both from $\lambda_1$ and $\lambda_2$. But if we split a subregion, then the entropy prescription would call for choosing the minimum between a surface that ends on brane 1 and a surface that ends on brane 2. The answer would only have one of the couplings, but the prescription would technically "see" both couplings so calling it a "local" prescription would be questionable.

---

## Round 1 · Referee Report · Anonymous (Referee 2) · 2023-9-2

Report

The article studies braneworld models in AdS where the theory that describes the brane contains, in addition to the brane tension, a DGP term. The aim is to obtain constraints on these brane parameters from the requirement that (i) the entanglement entropy on the brane is positive, and (b) that causal wedge inclusion is satisfied. Since these are constraints on the coefficients of the low-energy effective theory (LEFT) of the brane, they are regarded as similar to swampland constraints.

The idea of the article is new and interesting, and the article is well written, so I recommend its publication in SciPost after the points I raise are addressed.

Requested changes

1-In p.3, $\mathcal R$ appears without having defined it before.

2-In the last paragraph in p.4, it seems to me that (I) and (III) are interchanged relative to their definition two paragraphs above.

3-Positivity of entropy is justified in eq.(2.8) using the microcanonical concept of entropy as a coarse-graining over a number of microstates. However, the concept of entropy that is used in this paper is not this one but rather that of entanglement entropy, whose positivity follows from a different condition. In general, entanglement entropy is not directly related to a microstate counting, and even if in many circumstances such an interpretation turns out to be possible, this requires some elaboration. A reworking of the argument, in one direction or another, should be made in the paper.

4-Above eq.(2.12), $A_h$ and $S_h$ are referred to as distinct quantities (one upper-bounds the other) while in (2.12) they are set equal. This should be clarified.

5-The argument for CWI in black holes in p.12 is based on the claim that the $t=0$ section of a one-sided eternal black hole has no interior geometry. This does not seem correct to me. The section $t=0$ of the two-sided eternal black hole does not have any interior since the ER bridge connects two different exteriors, but in an eternal one-sided geometry, the ER bridge opens to a compact interior. Indeed it can contain an arbitrary interior (as in bag-of-gold examples). I would like the authors to clarify this point.

6-At the top of p.15 it says that with a $K^2$ term "a KR brane with only a redefined tension is not as obviously a solution." This sounds ambiguous. Is it never a solution, or can it be a solution in some cases? If it's the latter, then what argument can be used to exclude this term from the brane LE?

7-In my opinion fig.5 would be clearer if only non-redundant information were presented, ie the region below the diagonal, and only the strongest of the two constraints for cases of the same colour (the dashed line). This is a mere suggestion, and the authors may or not agree with it. Similarly for fig.6.

  • validity: -
  • significance: -
  • originality: -
  • clarity: -
  • formatting: -
  • grammar: -

Author:  Sanjit Shashi  on 2023-09-26  [id 4009]

(in reply to Report 2 on 2023-09-02)

We thank Referee 2 for their comments. Here are our responses to each of their points.

  1. We have have added the phrase "in a CFT subregion" before the first '$\mathcal{R}$.'

  2. We have corrected this typo.

  3. The main point we wanted to make was simply that von Neumann entropy must be positive. The statement about microstates was just meant to give an intuitive rationale, but we agree that it is a fallacious argument when discussing entanglement entropy. We have eliminated the references to the microstate argument throughout the document. Still, we can say that entanglement entropy looks like $\log \Omega$ for some $\Omega \geq 1$ by the Schmidt decomposition. $\Omega$ in this case is not a count (it need not be an integer), but we still have the crucial statement that entanglement entropy is positive. We have changed the presentation in the relevant part of Section 2.1 accordingly.

  4. Our point was phrased oddly in the previous version. The logical flow we want to express is that brane-localized couplings change the entropy functional, but the minimum (now defined as $S_{\text{min}}$) should be positive because it represents the entropy in the semiclassical limit. This is not necessarily computed by the horizon entropy $S_{\text{h}} \equiv A_{\text{h}}/(4G_{\text{N}})$, but we certainly have $S_{\text{h}} > S_{\text{min}}$. As $S_{\text{min}} > 0$, we deduce $S_{\text{h}} > 0$. We have modified the passage to better articulate this argument.

  5. Although we frequently said ``one-sided," we really had in mind a single side of a two-sided geometry. We have corrected this error throughout Section 2.2, including in the discussion of CWI.

  6. The referee's comment here refers to the v1 version on the arXiv. In the v2 version (which is also the SciPost v1), we had modified/corrected this statement. Specifically, we added a reference to arXiv:2206.06511, which found that adding a linear combination of $K^2$ and $K_{\mu\nu}K^{\mu\nu}$ to the action does not change the embedding equation. However, we clearly state that we remain uncertain how such terms might alter the entropy functional. Our only point here is that there are in principal other corrections of the same order in curvature that could be added to the action, but they might change some calculational details of the story.

  7. In Figure 5, we actually want to emphasize the overall $1 \leftrightarrow 2$ redundancy of the full set of constraints in the plot. As for Figure 6, because we have fixed $\theta_{1,2}$ in the numerics, there is actually no redundant information in the plots [except for 6(b) where the angles are the same], so excising the region either above of below the diagonal in those plots would remove information.

---

## Round 1 · Referee Report · Anonymous (Referee 3) · 2023-9-11

Report

In this paper the authors use consistency conditions on entropy to rule out certain values for DGP couplings in a particular bottom-up model of wedge holography. The paper is well written, clearly structured and suggests a nice way for how to link Quantum information theory, the Swapland programme and holography and I recommend its publication.

In addition to the other referees' comments I only have one point I would like to stress:

Requested changes

1- As already mentioned by another referee, the discussion of positivity of EE needs to be improved. In particular, 1. subregion entanglement entropy does not generally count microstates (as e.g. mentionted on p.3 or p.20). 2. positivity of EE in subregions is a scheme-dependent statement and relies on specifying a regularization and renormalization procedure (The statement about "typical AdS calculations" on p.20 is very vague and should be made more precise). 3. The fact that the brane construction gives UV finite answers should be discussed in a bit more detail. After all, since the region $\mathcal R$ under consideration is does not split a connected CFT region I would think that all the discussions about UV divergences are immaterial, since the authors work with a type I algebra whose EE is manifestly finite.

  • validity: -
  • significance: -
  • originality: -
  • clarity: -
  • formatting: -
  • grammar: -

Author:  Sanjit Shashi  on 2023-09-26  [id 4010]

(in reply to Report 3 on 2023-09-11)

We thank Referee 3 for their comments. Here are our responses to each of their points. As points 2 and 3 in particular are closely related, we do not number our responses.

First, as mentioned in our response to Referee 2, we have eliminated the phrasing of microstates in favor of the actual relevant statement---that the von Neumann entropy is guaranteed to be positive. The only time microstates are explicitly mentioned is specifically for microcanonical entropy in our exposition of the Boltzmann relation (2.8), and never for entanglement entropy.

Regarding the divergence of entanglement entropy and how positivity is scheme-dependent, we were trying to state that even though "bare" entanglement entropy diverges, it is still "positive." We of course agree that one can renormalize the entropy so as to eliminate the divergence and be left with a negative answer (depending on the scheme). However, our purpose in this exposition was to make it clear why such a negative renormalized entropy would not contradict positivity; we would say that positivity is only a requirement for the "bare" entropy.

Furthermore, we only mention "typical AdS calculations" (by which we mean no branes, meaning that RT surfaces generically pick up UV divergences since they reach the conformal boundary) to contrast again what is seen in the two-brane case. Of course, as mentioned by the referee, the key point needed in our work is that we get UV-finite entropies the two-brane case because we are not splitting subregions. Thus, we can get truly negative answers from the gravity side, even though we are not supposed to. Setting up the contrast between the two-brane and no-brane scenarios was really a decision about the framing of our story.

We have clarified these points in Section 2.1 under equation (2.8). We have also added more specificity to the paragraphs previously mentioning the "typical AdS calculations."

---

## Round 2 · Referee Report · Anonymous (Referee 2) · 2023-9-26

Report

In the revised version the authors have addressed my queries in a satisfactory manner. I recommend the publication of the article in its present form.

---

## Round 2 · Referee Report · Anonymous (Referee 1) · 2023-10-9

Report

The authors addressed all my observations in detail and in a satisfactory way. I am happy to recommend the publication of this paper on Scipost.

---

## Round 2 · Author Response

We thank the referees for their comments and questions. We have made several clarifications throughout the document. We have also corrected the issues pointed out across the reports.

---

## Round 2 · List of Changes

• Corrected typos
  • Sharpened the language around the presentation of results
  • Fixed the argument for entropy positivity
  • Added discussion/speculation on other higher-derivative terms
  • Clarified some of the highlighted vague points

---

## Editorial Decision

published